# Robust Sleep Staging over Incomplete Multimodal Physiological Signals via Contrastive Imagination

**Qi Shen**[1], **Junchang Xin**[2], **Bing Tian Dai**[3], **Shudi Zhang**[1], **Zhiqiong Wang**[1]*

[1]College of Medicine and Biological Information Engineering, Northeastern University, China
[2]College of Computer Science and Engineering, Northeastern University, China
[3]School of Information Systems, Singapore Management University, Singapore
2210521@stu.neu.edu.cn, xinjunchang@mail.neu.edu.cn, btdai@smu.edu.sg,
2310535@stu.neu.edu.cn, wangzq@bmie.neu.edu.cn

## Abstract

Multimodal physiological signals, such as EEG, EOG and EMG, provide rich and reliable physiological information for automated sleep staging (ASS). However, in the real world, the completeness of various modalities is difficult to guarantee, which seriously affects the performance of ASS based on multimodal learning. Furthermore, the exploration of temporal context information within PSs is also a serious challenge. To this end, we propose a robust multimodal sleep staging framework named **c**ontrastive **i**magination **m**odality **sleep net**work (CIMSleepNet). Specifically, CIMSleepNet handles the issue of arbitrary modal missing through the combination of modal awareness imagination module (MAIM) and semantic & modal calibration contrastive learning (SMCCL). Among them, MAIM can capture the interaction among modalities by learning the shared representation distribution of all modalities. Meanwhile, SMCCL introduces prior information of semantics and modalities to check semantic consistency while maintaining the uniqueness of each modality. Utilizing the calibration of SMCCL, the data distribution recovered by MAIM is aligned with the real data distribution. We further design a multi-level cross-branch temporal attention mechanism, which can facilitate the mining of cross-scale temporal context representations at both the intra-epoch and inter-epoch levels. Extensive experiments on five multimodal sleep datasets demonstrate that CIMSleepNet remarkably outperforms other competitive methods under various missing modality patterns. The source code is available at: https://github.com/SQAIYY/CIMSleepNet.

## 1 Introduction

Automated sleep staging (ASS) is essential to promote sleep quality assessment and sleep disorder diagnosis, providing convenience for the public in the daily monitoring of sleep within their home environment. Many machine learning algorithms, including feature engineering and deep learning, have been proposed for ASS [1, 2, 3, 4, 5]. In particular, deep learning methods represented by convolutional neural network (CNN) have achieved remarkable results in the field of ASS [6]. Compared with feature engineering, deep learning does not require the guidance of prior knowledge and has the advantage of automatically extracting physiological signals (PSs) features.

In clinical applications, due to the complexity of human physiological states, subjects usually need to wear multiple sensors to obtain more comprehensive and integrated physiological information from multimodal PSs collected from different sources [7]. Hence, several multimodal fusion algorithms

---

*Corresponding author

38th Conference on Neural Information Processing Systems (NeurIPS 2024).

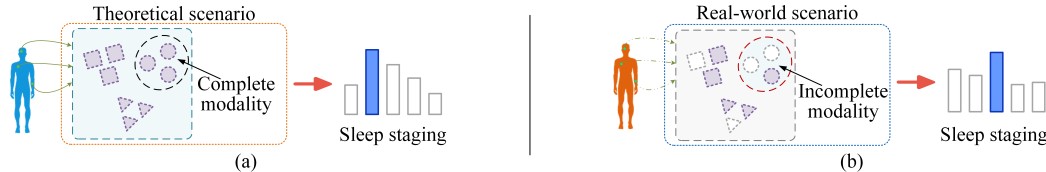

Figure 1: The distribution of multimodal data in different scenarios. (a) exhibits the complete modality, and (b) exhibits the incomplete modality.

[8, 9, 10, 11] based on deep learning have been developed to cope with the challenges of multimodal ASS. Although various multimodal fusion algorithms provide guarantees for automated processing and analysis of these multimodal PSs, they still have some limitations. As illustrated in the Fig. 1 (a), existing methods are almost all conducted under the assumption that all modal data are complete. However, in real scenarios, the modal data will be incomplete due to sensor malfunctions or detachment, as shown in the Fig. 1 (b). Unfortunately, the second scenario will seriously affect the reasoning process of algorithms, resulting in a sharp decline in performance [12].

Further, how to mine dynamic temporal changes and complex stage-transitioning patterns in PSs is another challenge for ASS. Most sleep staging works [13, 14, 15, 16, 17] utilize recurrent neural network (RNN) and its variants to model temporal dependencies within learnable hidden states. Recently, due to its efficient parallel computing ability and powerful global context modeling ability, Transformer has gradually become the preferred alternative to RNN in the ASS field [18, 19, 20]. However, Transformer lacks the recurrent modeling abilities of RNN, which is crucial for mining the structural representations and positional embedding of input sequences [21, 22]. Meanwhile, most methods are limited to mining temporal correlations at a single level in PSs, i.e., intra-epoch level or inter-epoch level. These issues make it difficult for existing temporal models to fully understand the complex variability patterns in PTS, thereby affecting the performance of sleep staging.

Considering the above challenges, we propose a robust multimodal sleep staging framework named **c**ontrastive **i**magination **m**odality **sleep net**work (CIMSleepNet), suitable for scenarios with incomplete modalities. The core contributions of CIMSleepNet are summarized as follows.

- We first design a modal awareness imagination module (MAIM), which can realize the imputation of missing modalities to restore the completeness of the various modalities. MAIM leverages the distribution of available modalities as prior conditions to learn multimodal shared representations and enhance the inter-modal correlation, thereby improving the recovery process of missing modalities.
- We provide a novel insight into the impact of the intrinsic connection between semantic and modality on data distribution. Hence, a semantic & modal calibration contrastive learning (SMCCL) is presented to modify the restored data distribution. It can utilize bidirectional guidance of semantic and modality to align the restored data with the real distribution.
- We further explore a multi-level cross-branch temporal attention (MCTA) mechanism that enables interactive modeling of recurrent features and self-attention weights from the intra-epoch and inter-epoch levels to yield more comprehensive temporal representations.
- Extensive experiments on five multimodal sleep datasets exhibit that CIMSleepNet can significantly improve multimodal ASS performance under various missing modality patterns.

## 2 Related Work

**Multimodal Learning for Sleep Staging**: In the ASS field, several pioneering studies have been devoted to exploring how to utilize multimodal PSs acquired from various sensors to improve ASS performance. Andreotti et al. [8] selected three polysomnography (PSG) signals related to sleep, electroencephalogram (EEG), electrooculogram (EOG) and electromyogram (EMG), as input to CNN to improve ASS accuracy. Similarly, Jia et al. [23] effectively mined salient waves form multimodal sleep PSs with a multimodal salient wave detection network. Lin et al. [11] designed a cross-link fusion module to eliminate redundant information in multimodal PSs. Huy et al. [8] focused on the training mode of the deep model, and proposed an adaptive gradient blending strategy, which

improves the joint learning representation ability of multimodal PSs in different views. Furthermore, multimodal PSs collected by some consumer electronic devices have gradually been applied in ASS field. For instance, Walch et al. [24] utilized feature engineering methods to analyze human motion signals and heart rate (HR) signals collected by Apple Watch, and verified their relevance to the sleep stage. Then, Zhai et al. [9] and Mads et al. [25] further improved multimodal sleep staging performance based on consumer electronic devices by constructing a feature fusion method based on deep learning. However, these studies have largely neglected the impact of incomplete modalities scenarios, which are more representative of real-world data distributions. Kontras et al. [26] ingeniously combined self-attention and cross-attention mechanisms to extract coordinating representations for multimodal PSs, thereby mitigating the interference caused by missing modalities on neural network. Nevertheless, this method was developed to handle the complete absence of one or more modalities, whereas it is impractical in real-life clinical applications.

**Contrastive Learning Under Missing Modalities**: Invariant contrastive learning (ICL) and semantic contrastive learning (SCL) are currently promising choices for solving the modality missing issue. For instance, Lin et al. [27] proposed a cross-modal ICL, aiming to utilize available modalities to achieve prediction of missing modalities. Similarly, Liu et al. [28] narrowed the gap between heterogeneous modalities through ICL for reconstructing missing modalities. SCL introduces category information on the basis of the former to achieve semantic structure preservation in missing modal cases [29, 30]. These studies focus on learning multimodal consistency representations, i.e., only recovering the multimodal shared information to deal with multimodal missing issues. However, this strategy leads to the loss of specific information unique to each modality, thereby failing to exploit inter-modal complementarity.

**Temporal Context Learning in sequence modeling**: It has achieved rapid development driven by the sequence-to-sequence models. For instance, Supratak et al. [13] introduced bidirectional long short-term memory (Bi-LSTM) to learn transition rules during sleep stages. Phan et al. [14] applied bidirectional gated recurrent unit (Bi-GRU) to model contextual information of sequence representations. Phyo et al. [16] provided a Bi-LSTM equipped with two auxiliary tasks to explicitly learn periodic transition patterns. Besides, Qu et al. [18] employed Transformer to improve the ability to mine context information in a parallel optimization manner. Eldele et al. [19] deployed temporal CNN to Transformer, further improving its ability to capture temporal features. Although Transformer has advantages over RNN and its variants in terms of computational efficiency and context learning, it lacks recurrent modeling ability, resulting in the omission of some important temporal attribute information [21, 22]. Furthermore, studies [22, 31, 32, 33] have proved that the features learned by RNN and Transformer are complementary. The above optimization perspective provides valuable inspiration for us to design novel temporal context architectures.

## 3 Methodology

### 3.1 Problem Formulation

We first define a complete multimodal PSs dataset $\mathbb{D} = \{(\mathbf{X}_i, y_i)\}_{i=1}^N$ where $\mathbf{X}_i$ is the $i$th multimodal epoch (sample), $y_i$ is the sleep stage label of the $i$th epoch and $N$ is total number of epochs. Suppose $\mathbf{X}_i$ contains $M$ modities, i.e., $\mathbf{X}_i = \{\mathbf{x}_i^j\}_{j=1}^M$, $\mathbf{x}_i^j \in \mathbb{R}^{C_j \times L_j}$, where $C_j$ and $L_j$ are the number of channels and sampling points of the $j$th modality, respectively. Furthermore, $y_i \in \{0, 1, \cdots, K-1\}$, where $K$ is the number of sleep stage categories. Different from the complete modality missing issue of Kontras et al. [26], we mainly focus on the chunk-based missing pattern, i.e., random missing in units of multiple epochs, which is a common situation in biomedical research [34]. This is mainly due to the fact that subjects tend to be interrupted for an extended period of time during the data collection. To construct incomplete modal dataset, we define a mask matrix $\mathbf{Z} = \{\{Z_i^j\}_{i=1}^N\}_{j=1}^M \in \mathbb{R}^{N \times M}$ at the epoch level to track the missing status of modalities. If $\mathbf{x}_i^j$ is observed, $Z_i^j = 1$; otherwise, $Z_i^j = 0$. Note that, $Z_i^0 \wedge Z_i^1 \wedge, \cdots, \wedge Z_i^{M-1} \neq 0$, i.e., each $\mathbf{X}_i$ must have at least one available modality. According to the mask matrix, the missing rate of the dataset can be defined as $\rho = 1 - \frac{1}{N \cdot M} \sum_{i=1}^N \sum_{j=1}^M Z_i^j$. Then, we define the incomplete multimodal PSs dataset $\tilde{\mathbb{D}} = \{(\tilde{\mathbf{X}}_i, y_i)\}_{i=1}^N$, where $\tilde{\mathbf{X}}_i$ and $\mathbf{X}_i$ have the same shape, i.e., $\tilde{\mathbf{X}}_i = \{\tilde{\mathbf{x}}_i^j\}_{j=1}^M$, $\tilde{\mathbf{x}}_i^j \in \mathbb{R}^{C_j \times L_j}$. After that, we reorganize the dataset $\tilde{\mathbb{D}}$ with a new shape, i.e., $\mathbf{x}^j \in \mathbb{R}^{\vec{N} \times T \times C_j \times L_j}$, to perform temporal context modeling. Among them, $\vec{N} = \lfloor N/T \rfloor$ and $T$ is the length of contextual information. Finally,

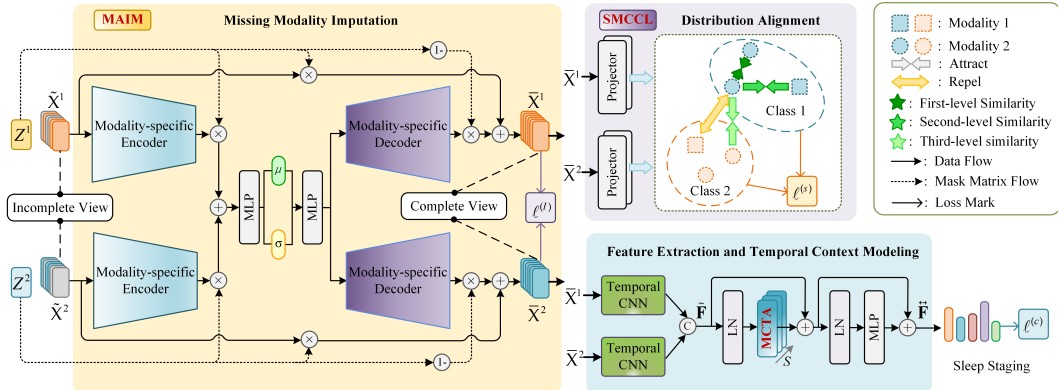

Figure 2: The overall framework of CMISleepNet. It consists of three main components: MAIM, SMCCL and MCTA mechanism. Two incomplete modalities, $\tilde{\mathbf{X}}^1$ and $\tilde{\mathbf{X}}^2$ are taken as examples for illustration. In the missing modality imputation phase, MAIM learns multimodal shared representations from the available modal distribution to recover complete modalities $\bar{\mathbf{X}}^1$ and $\bar{\mathbf{X}}^2$. Meanwhile, $\bar{\mathbf{X}}^1$ and $\bar{\mathbf{X}}^2$ are fed into SMCCL to perform distribution alignment, making the recovered modal data closer to the real data distribution. Furthermore, temporal CNN is utilized to performer feature extraction of $\bar{\mathbf{X}}^1$ and $\bar{\mathbf{X}}^2$ and obtain the multimodal fusion representation $\tilde{\mathbf{F}}$. After that, $\tilde{\mathbf{F}}$ is fed into a Transformer containing MCTA for temporal context modeling to obtain the temporal representation $\vec{\mathbf{F}}$, which is then used for prediction of sleep stage scores. CMISleepNet also includes three objective functions: $\ell^{(I)}$ for missing modality imputation, $\ell^{(s)}$ for distribution alignment, $\ell^{(c)}$ for sleep staging.

we also define a modality matrix $\mathbf{S} = \{\, \{\, s_i^j \,\}_{i=1}^{H}\, \}_{j=1}^{M} \in R^{H \times M}$ to provide information about the modalities involved in each epoch, where $s_i^j \in \{0, 1, \cdots, M-1\}$ is the modal label of the $j$th modality of the $i$th epoch and $H = \vec{N} \cdot T$.

As schematized in Fig. 2, we present CIMSleepNet, which aims to cope with the issues of modality missing and temporal context modeling in multimodal ASS. Given incomplete multimodal PSs, we first employ MAIM to impute the missing modal data (Sec. 3.2). Meanwhile, SMCCL is utilized to modify the distribution of the recovered data (Sec. 3.3). Then, we leverage temporal CNN and MCTA embedded in the Transformer structure to perform feature extraction and temporal context modeling on the recovered complete multimodal data, respectively (Sec. 3.4). Finally, the model parameters are optimized by combining various objective functions to achieve sleep staging (Sec. 3.5).

## 3.2 Missing Modality Imputation

To impute missing modalities, we design MAIM, which mainly consists of $M$ ($M$=2 in Fig. 2) modality-specific encoders $\mathbf{E}(\cdot) = \{E_j(\cdot)\}_{j=1}^{M}$ and decoders $\mathbf{D}(\cdot) = \{D_j(\cdot)\}_{j=1}^{M}$. Each encoder and decoder is implemented via separable temporal CNN [16] to reduce the parameter redundancy.

**Modality-specific encoder.** As as depicted in Fig. 2, incomplete multimodal PSs $\tilde{\mathbf{X}}$ and mask matrix $\mathbf{Z}$ are transmitted in MAIM through multimodal data flow and mask matrix flow respectively. Firstly, multiple encoders are utilized to project multimodal PSs into the latent space, and the latent representations of all modalities are fused in a multiply add operation, formulated as

$$\mathbf{f}_i = \frac{1}{\sum_{j=1}^{M} Z_i^j} \sum_{j=1}^{N} Z_i^j E_j\left(\tilde{\mathbf{x}}_i^j\right) \tag{1}$$

where $\mathbf{f}_i$ denotes the multimodal shared representation obtained from $\mathbf{X}_i$. Since the modalities in the training set are also incomplete, the best choice for guiding the missing data in the reconstruction process is other available data of the same modality [30]. However, the data recovered by this way loses the diversity of the original data and cannot retain the original semantic structure. To improve the data diversity, we drew inspiration from multimodal variational autoencoder (MVAE) [35] to learn not the shared representations of multimodal PSs but their distributions. Learning diverse data ensures that the generated data is not limited to the data that guide it, making it easier for SMCCL to perform calibration. We first utilize multilayer perceptron (MLP) to obtain two vectors, $\mu_i$ and $\sigma_i$, which are used to describe the mean and variance in the distribution from the $\mathbf{f}_i$. Then, $\mathbf{f}_i$ is subjected

to reparameterization to obtain the latent representation $\hat{\mathbf{f}}_i$. Formally, $\hat{\mathbf{f}}_i = \mu_i + \exp\left(\frac{\sigma_i}{2}\right) \odot \varepsilon_i$, where $\odot$ is element-wise multiplication and $\varepsilon_i$ is a random variable sampled from the distribution of $\mathbf{f}_i$. After that, $\hat{\mathbf{f}}_i$ is mapped back into the input space of $\mathbf{f}_i$ to get multimodal shared representation $\bar{\mathbf{f}}_i$.

**Modality-specific decoder.** In the decoding stage, $\bar{\mathbf{f}}_i$ is fed into each decoder for reconstructing modality-specific data, i.e., $\{\bar{\mathbf{x}}_i^j\}_{j=1}^M = \{D_j(\bar{\mathbf{f}}_i)\}_{j=1}^M$. Similar to MVAE, the parameters of MAIM are optimized guided by the joint of mean square error (MSE) $\ell^{(mse)}$ and Kullback-Leibler (KL) divergence $\ell^{(KL)}$. We refer to the overall loss function as the modal imagination loss function $\ell^{(I)}$. Suppose the batchsize is $h$, $\ell^{(I)}$ can be denoted as

$$
\begin{aligned}
\ell^{(I)} &= \frac{1}{M}\sum\nolimits_j^M \ell_j^{(mse)} + \eta\ell^{(KL)} \\
&= \frac{1}{M\cdot B}\sum\nolimits_{j=1}^M\sum\nolimits_{i=1}^B \left\|\tilde{\mathbf{x}}_i^j - \bar{\mathbf{x}}_i^j\right\|^2 - \frac{\eta}{2B}\sum\nolimits_{i=1}^B\sum\nolimits_{k=1}^{\bar{D}}\left(1 + \ln\left(\sigma_i^k\right) - \left(\mu_i^k\right)^2 - \left(\sigma_i^k\right)^2\right)
\end{aligned}
\tag{2}
$$

where $B = h \cdot T$, $\bar{D}$ is the dimension of $\bar{\mathbf{f}}_i$, $\eta$ is the loss weight, $\tilde{\mathbf{x}}_i^j$ is the real sample (if $\tilde{\mathbf{x}}_i^j$ is missing, $\tilde{\mathbf{x}}_i^j$ is the random sampling of the available data in the same modality). We found that the value of $\eta$ is not sensitive, but removing $\ell^{(KL)}$ results in a significant decrease in performance of CMISleepNet. Hence, we set $\eta$ to 1. In particular, $\ell^{(KL)}$ is used to constrain how close the latent variable distribution is to the prior distribution, prompting the decoder to generate more diverse samples. Then, the mask matrix is utilized to judge whether all recovered data is in a missing state before. if $\tilde{\mathbf{x}}_i^j$ is missing, $\bar{\mathbf{x}}_i^j$ will be used as the recovered modality; otherwise, $\tilde{\mathbf{x}}_i^j$ itself will be used. It can be expressed by mask matrix as $\bar{\mathbf{x}}_i^j = Z_i^j\tilde{\mathbf{x}}_i^j + (1 - Z_i^j)\bar{\mathbf{x}}_i^j$.

### 3.3 Distribution Alignment

Different from contrastive learning based on modality consistency [27, 28, 30, 29], our SMCCL introduces semantic and modal information, which not only preserves the semantic structure but also restores the specific modality information to a great extent. As illustrated in Fig. 2, SMCCL covers three similarity levels. The first-level similarity is applied to narrow the distance between different samples with two identical patterns, i.e., the same category and the same modality. Second-level and third-level similarities are utilized to correct the distribution between samples with any of the same single patterns. Note that, the constraint strength of the first-level similarity should be higher than that of the other two levels of similarity because it can be dual-guided in semantics and modality. The latter two levels of similarity are meaningful, and samples that meet these similarity criteria should not be repelled. Because these data still have semantic similarity or modal similarity. Furthermore, contrastive learning is performed within a batch, and the original complete data that meets the first-level similarity standard with the restored data may not necessarily exist in a batch, which further reflects the necessity of the latter two levels of similarity.

Supposing that a batch contains $B$ epochs, we divide the above similarity levels by constructing similarity weight matrix $\mathbf{W} = \{\{w_i^j\}_{i=1}^{B\times M}\}_{j=1}^{B\times M}$. To divide the similarity levels of all sample pairs, we use the label set $\{y_i\}_{i=1}^B$ and modality matrix $\mathbf{S}$ to introduce both semantic and modal information for each sample. We first replicate the label set, increasing its modality dimension, to obtain label weight $\mathbf{Y} = \{\{\tilde{y}_i^j\}_{i=1}^B\}_{j=1}^M$. Flatten two matrices and replicate in the row and column dimension to expand to $R = B \cdot M$. We redefine two matrices as $\bar{\mathbf{Y}} = \{\{\bar{y}_i^j\}_{i=1}^R\}_{j=1}^R$ and $\bar{\mathbf{S}} = \{\{\bar{s}_i^j\}_{i=1}^R\}_{j=1}^R$. Then, calculate the contrastive mask matrices of $\bar{\mathbf{Y}}$ and $\bar{\mathbf{S}}$, $\mathbf{U}$ and $\mathbf{V}$, formulated as:

$$
\mathbf{U} = \{\{u_i^j\}_{i=1}^R\}_{j=1}^R, u_i^j = \left\{\begin{array}{ll} 1, & \bar{y}_i^j = \dot{y}_i^j \\ 0, & \bar{y}_i^j \neq \dot{y}_i^j \end{array}\right. \quad \mathbf{V} = \{\{v_i^j\}_{i=1}^R\}_{j=1}^R, v_i^j = \left\{\begin{array}{ll} 1, & \bar{s}_i^j = \dot{s}_i^j \\ 0, & \bar{s}_i^j \neq \dot{s}_i^j \end{array}\right. \tag{3}
$$

where "1" is a positive pair and "0" is a negative pair. Besides, $\dot{y}_i^j$ and $\dot{s}_i^j$ are the elements in $\bar{\mathbf{Y}}^T$ and $\bar{\mathbf{S}}^T$ respectively. Further, the similarity weight matrix $\mathbf{W}$ can be constructed by

$$
\mathbf{W} = \underbrace{\mathbf{U}\odot\mathbf{V}}_{\text{the 1th level}} + \underbrace{(1\text{-}\boldsymbol{\Theta})(\mathbf{U} - \mathbf{U}\odot\mathbf{V})}_{\text{the 2th level}} + \underbrace{\boldsymbol{\Theta}(\mathbf{V} - \mathbf{U}\odot\mathbf{V})}_{\text{the 3th level}} \tag{4}
$$

where $\odot$ denotes element-wise multiplication and $\boldsymbol{\Theta} = \{\{\Theta_i^j\}_{i=1}^R\}_{j=1}^R$ is used to set the weights for the second-level and third-level similarity. We refer to $\boldsymbol{\Theta}$ as the modality consistency matrix and $\Theta_i^j$ as the modality consistency score. In $\boldsymbol{\Theta}$, $\{\{\Theta_i^j\}_{i=(k-1)\cdot B+1}^{k\cdot B}\}_{j=1}^R$ is the modality consistency score of the $k$th modality and other modalities, which contain all the same $\Theta_i^j$ values. We rename $\Theta_i^j$ in $\{\{\Theta_i^j\}_{i=(k-1)\cdot B+1}^{k\cdot B}\}_{j=1}^R$ to $\theta_k$ and calculate it by the inter-modal mutual information under

information theory [36]. Taking the $k$th modality as an example, we use the projector $g_k(\cdot)$ composed of MLP to map the reconstructed complete modality data into a low-dimensional feature space and activate it by the Softmax function $\delta_k(\cdot)$, i.e., $\phi^k = \delta_k(g_k(\bar{x}^k))$. Formally,

$$\theta_k = \frac{1}{M-1} \sum\nolimits_{i=1}^{M} \mathbb{1}_{i \neq k} \cdot \frac{I(\phi^k; \phi^i)}{H(\phi^k, \phi^i)} \tag{5}$$

where $\mathbb{1}_x$ is an indicator, when $x$ is true, the result is "1", otherwise it is "0", $I(\phi^k; \phi^i)$ is the mutual information of $\phi^k$ and $\phi^i$, $H(\phi^k, \phi^i)$ is the joint entropy of $\phi^k$ and $\phi^i$. The value range of $\theta_k$ is between 0 and 1, and it can automatically adjust the ratio of the second-level and third-level similarity according to the modal consistency. For instance, if the value of $\theta_k$ is larger, it means that the inter-modal consistency is higher, but the amount of specific modal information is lower. Hence, it is necessary to increase the introduction of modal information, i.e., to increase the weight of the third-level similarity of formula (4). Vice versa. To more intuitively represent the construction process of the similarity weight $\mathbf{W}$, we provide an example in Appendix C. To formulate $\frac{I(\phi^k; \phi^i)}{H(\phi^k, \phi^i)}$, we define a discrete joint probability distribution $\mathcal{P}(m, n)$ and two discrete marginal probability distributions $\mathcal{P}(m)$ and $\mathcal{P}(n)$. Since $\phi^k$ and $\phi^i$ are activated by Softmax function, $\phi^k$ and $\phi^i$ can be regarded as the distribution of two discrete cluster assignment variables $m$ and $n$ on $\dot{D}$ categories like [29, 37]. Among them, $\dot{D}$ is the feature dimension of $\phi^k$ and $\phi^i$. Hence, we redefine $\mathcal{P}(m, n)$, $\mathcal{P}(m)$ and $\mathcal{P}(n)$ as $\mathbf{P} = \frac{1}{2}(\phi^k(\phi^i)^{\mathsf{T}} + \phi^i(\phi^k)^{\mathsf{T}}) \in \mathbb{R}^{B \times \dot{D} \times \dot{D}}$, $\mathbf{P}_m = \text{Expand}(\frac{1}{\dot{D}} \sum_{d_n=1}^{\dot{D}} \mathbf{P}_{\bullet, \bullet, d_n}) \in \mathbb{R}^{B \times \dot{D} \times \dot{D}}$ and $\mathbf{P}_n = \text{Expand}(\frac{1}{\dot{D}} \sum_{d_m=1}^{\dot{D}} \mathbf{P}_{\bullet, d_m, \bullet}) \in \mathbb{R}^{B \times \dot{D} \times \dot{D}}$, respectively. As a result, the discrete form of $\frac{I(\phi^k; \phi^i)}{H(\phi^k, \phi^i)}$ can be expressed as

$$\frac{I(\phi^k; \phi^i)}{H(\phi^k, \phi^i)} = \log_{\frac{1}{\mathbf{P}}} \left( \frac{\mathbf{P}}{\mathbf{P}_m \mathbf{P}_n} \right) \tag{6}$$

The theoretical result of formula (6) are demonstrated in Appendix D. To match the dimensions of the two redefined matrices $\bar{\mathbf{Y}}$ and $\bar{\mathbf{S}}$, we perform a flatten operation on each batch of reconstructed data to obtain $\dot{\mathbf{X}} = \{\bar{x}_i\}_{i=1}^{B \cdot M}$. Then, we fed $\dot{\mathbf{X}}$ into another projector $\bar{g}(\cdot)$ for the computation of contrastive loss, i.e., $\psi = \bar{g}(\dot{\mathbf{X}}), \psi = \{\varphi_i\}_{i=1}^{B \cdot M}$. According to $\mathbf{W}$, we propose a novel contrastive learning, SMCCL, which can be defined as

$$\ell^{(s)} = \frac{-1}{N_{w_i^j > 0} - 1} \sum_{i=1}^{B \cdot M} \sum_{j=1}^{B \cdot M} \mathbb{1}_{i \neq j} \cdot \mathbb{1}_{w_i^j > 0} \cdot w_i^j \cdot \log \frac{\exp(\varphi_i \cdot \varphi_j / \tau)}{\sum_{k=1}^{B \cdot M} \mathbb{1}_{i \neq k} \cdot \exp(\varphi_i \cdot \varphi_k / \tau)} \tag{7}$$

where $\ell^{(s)}$ is named distribution alignment loss, $N_{w_i^j > 0} - 1$ is the number of $w_i^j > 0$ in a batch and $\tau$ is a temperature coefficient, which is set to 0.07 like [38]. In SMCCL, $\ell^{(s)}$ adjusts the attention given to different sample pairs based on $\mathbf{W}$, achieving more fine-grained distribution calibration.

## 3.4 Feature Extraction and Temporal Context Modeling

As illustrated in Fig. 2, the recovered complete modal dataset $\bar{\mathbf{X}} = \{\{x_i^j\}_{i=1}^{B}\}_j^{M}$ is also fed into the temporal CNN for feature extraction and concatenation to obtain multimodal fusion temporal representation $\tilde{\mathbf{F}} \in \mathbb{R}^{B \times D \times C}$, $B = h \cdot T$ during the distribution calibration process. Among them, $C$ is the number of channels, $D$ is the feature dimension, $h$ is the batch size and $T$ is the context length. Then, we utilize a Transformer composed of layer normalization (LN), MCTA, and MLP for temporal context modeling, thereby obtaining temporal representation $\ddot{\mathbf{F}} \in \mathbb{R}^{B \times D \times C}$. We focus on introducing MCTA, with its single-head structure depicted in Fig. 3. Firstly, the fusion representation after the first LN is divided into $S$ heads, i.e., $\dot{\mathbf{F}} = \{\dot{f}_s\}_{s=1}^{S}$, where $\dot{f}_s \in \mathbb{R}^{B \times D \times (C/S)}$. After that, $\dot{f}_s$ is fed into MCTA. It has two branches and includes intra-epoch and inter-epoch levels, which can fully mine the temporal context information of latent features.

**Intra-epoch level**: In the 1th branch, temporal CNN is adopted to generate the query $Q_s$ and the key $K_s^{(T)}$ and value $V_s^{(T)}$. Related study [39] have proven that temporal CNN exhibits efficiency beyond linear operations, while also eliminating the requirement for positional encoding. In the 2th branch, we use Bi-GRU to learn the recurrent representation of $\dot{f}_s$. Similarly, key $\bar{K}_s^{(B)}$ and value $\bar{V}_s^{(B)}$ from $\bar{\mathbf{f}}_s^{(B)}$ are obtained via temporal CNN. To achieve cross-branch interaction, we splice $K_s^{(T)}$ and $V_s^{(T)}$ with $\bar{K}_s^{(B)}$ and $\bar{V}_s^{(B)}$. As a result, the intra-epoch cross-branch attention can be calculated as

$$\dot{\mathbf{f}}_s^{(T)} = \text{Intra\_CA}_s = \text{Softmax}(\frac{Q_s \cdot K_s^{\mathsf{T}}}{\sqrt{C/S}})V_s, K_s = \left[K_s^{(T)} || \bar{K}_s^{(B)}\right], V_s = \left[V_s^{(T)} || \bar{V}_s^{(B)}\right] \tag{8}$$

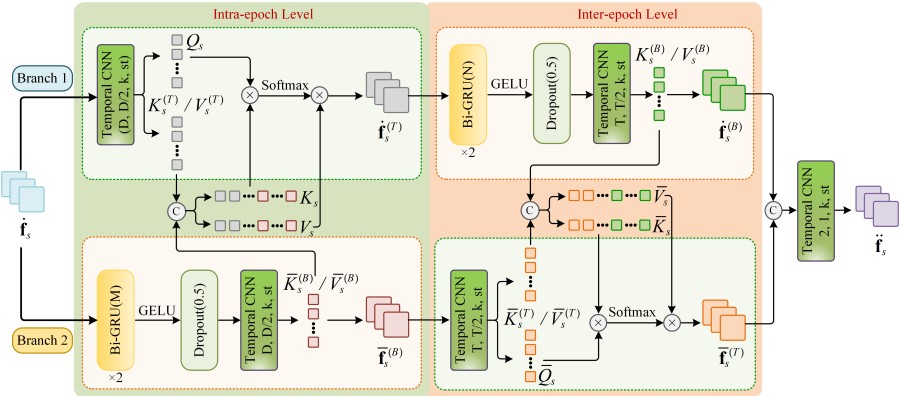

Figure 3: Design of the multi-level cross-branch temporal attention (MCTA) mechanism. $D$ and $T$ are the number of channels of temporal CNN at different levels; the values of $D/2$ and $T/2$ are rounded down; k is the kernel size; st is the stride. $M$ and $N$ are the neuron counts of Bi-GRU at different levels, where $M = C/S$ and $N = D \cdot C/S$.

In the interactive process, MCTA can effectively integrate recurrent bias into self-attention weights to improve the shortcomings of traditional Transformer recurrent modeling ability.

**Inter-epoch level**: As shown in Fig. 3, the 1th branch and the 2th branch of the inter-epoch level exhibit a reversed pattern compared to the intra-epoch level. This design enables MCTA to not only realize the interaction of cross-branch in parallel manner, but also capture rich temporal representations layer by layer. In this level, $\dot{\mathbf{f}}_s^{(T)} \in \mathbb{R}^{h \times T \times (D \cdot C/S)}$ and $\overline{\mathbf{f}}_s^{(B)} \in \mathbb{R}^{h \times T \times (D \cdot C/S)}$ serve as the input of the two branches respectively. In the 1th branch, similar to the 2th branch at the intra-epoch level, $\dot{\mathbf{f}}_s^{(T)}$ is mapped to $\dot{\mathbf{f}}_s^{(B)}$, to obtain $K_s^{(B)}$ and $V_s^{(B)}$. In the 2th branch, $\overline{\mathbf{f}}_s^{(B)}$ is mapped to $\bar{Q}_s$, $\bar{K}_s$ and $\bar{V}_s$ by temporal CNN. Likewise, the inter-epoch cross-branch attention can be calculated as

$$\overline{\mathbf{f}}_s^{(T)} = \text{Inter\_CA}_s = \text{Softmax}(\frac{\bar{Q}_s \cdot \bar{K}_s^{\mathrm{T}}}{\sqrt{C/S}})\bar{V}_s, \bar{K}_s = \left[\bar{K}_s^{(T)}||K_s^{(B)}\right], V_s = \left[\bar{V}_s^{(T)}||V_s^{(B)}\right] \quad (9)$$

After that, we concatenate $\dot{\mathbf{f}}_s^{(B)} \in \mathbb{R}^{h \times T \times (D \cdot C/S)}$ and $\overline{\mathbf{f}}_s^{(T)} \in \mathbb{R}^{h \times T \times (D \cdot C/S)}$, and perform dimensionality reduction via temporal CNN to obtain the fused representation $\ddot{\mathbf{f}}_s \in \mathbb{R}^{B \times (D/S)}$. Finally, extending single-head MCTA to multiple heads can be expressed as $\ddot{\mathbf{F}} = [\ddot{\mathbf{f}}_1||\ddot{\mathbf{f}}_2||\cdots||\ddot{\mathbf{f}}_S] \in \mathbb{R}^{B \times \ddot{D}}$.

### 3.5 Optimization Objective

We utilize temporal representation $\overleftrightarrow{\mathbf{F}} \in \mathbb{R}^{B \times D \times C}$ to perform sleep staging. Meanwhile, cross entropy loss $\ell^{(c)}$ is regarded as a good choice to guide the learning of model parameters, i.e.,

$$\ell^{(c)} = -\frac{1}{B} \sum_{i=1}^{B} \sum_{j=1}^{K} \tilde{\mathbf{W}}_j \left(y_{i,j} \ln\left(\tilde{y}_{i,j}\right) + (1 - y_{i,j}) \ln\left(1 - \tilde{y}_{i,j}\right)\right) \quad (10)$$

where $B$ is the batch size, $K$ is the number of categories, $\tilde{\mathbf{W}}$ is the category weight, $y$ is the real label and $\tilde{y}$ is the predicted label. After that, we construct the total objective loss for CIMSleepNet. Formally, $\ell = \ell^{(c)} + \alpha\ell^{(I)} + \beta\ell^{(s)}$, where $\alpha$ and $\beta$ are the weight of the loss term.

## 4 EXPERIMENTS

### 4.1 Datasets and Implementation Details

**Datasets**: Five multimodal sleep datasets, Sleep-EDF-20 [40, 41], Sleep-EDF-78 [40, 41], SVUH-UCD [40], Motion and heart rate (MHR) [24] and SHHS [42, 43] are used for the effectiveness of CIMSleepNet. The first four datasets are used to verify the performance of CIMSleepNet when the modality is **randomly partially missing**, and the last dataset is used to verify its performance when the modality is **completely missing**. We choose EEG and EOG, for Sleep-EDF-20, Sleep-EDF-78

Table 1: Performance comparison for complete and incomplete modalities in randomly partially missing case. Here "incomplete" means the maximum missing rate.

| Datasets | Methods | Complete | | | Incomplete | | |
|---|---|---|---|---|---|---|---|
| | | Acc | MF1 | $K$ | Acc | MF1 | $K$ |
| Sleep-EDF-20 | FeatConcat | 0.825 | 0.761 | 0.771 | 0.497 | 0.429 | 0.285 |
| | MultitaskCNN [8] | 0.835 | 0.753 | 0.775 | 0.589 | 0.506 | 0.449 |
| | SalientSleepNet [23] | **0.872** | **0.827** | **0.827** | 0.634 | 0.565 | 0.485 |
| | MM-Net [11] | 0.867 | 0.817 | 0.822 | 0.570 | 0.493 | 0.432 |
| | TransSleep [16] | 0.864 | 0.819 | 0.821 | 0.594 | 0.521 | 0.457 |
| | XSleepNet [10] | 0.864 | 0.809 | 0.819 | 0.623 | 0.560 | 0.478 |
| | CIMSleepNet | 0.867 | 0.821 | 0.824 | **0.853** | **0.801** | **0.805** |
| Sleep-EDF-78 | FeatConcat | 0.788 | 0.726 | 0.717 | 0.526 | 0.471 | 0.392 |
| | MultitaskCNN [8] | 0.795 | 0.727 | 0.722 | 0.613 | 0.535 | 0.453 |
| | SalientSleepNet [23] | 0.843 | 0.794 | 0.791 | 0.722 | 0.643 | 0.625 |
| | MM-Net [11] | 0.845 | 0.796 | 0.794 | 0.706 | 0.628 | 0.597 |
| | TransSleep [16] | 0.846 | 0.797 | 0.795 | 0.738 | 0.654 | 0.637 |
| | XSleepNet [10] | 0.838 | 0.776 | 0.779 | 0.697 | 0.622 | 0.583 |
| | CIMSleepNet | **0.849** | **0.799** | **0.797** | **0.830** | **0.772** | **0.775** |
| SVUH-UCD | FeatConcat | 0.745 | 0.731 | 0.672 | 0.502 | 0.445 | 0.336 |
| | MultitaskCNN [8] | 0.774 | 0.763 | 0.705 | 0.643 | 0.630 | 0.533 |
| | TransSleep [16] | 0.794 | 0.782 | 0.732 | 0.725 | 0.698 | 0.636 |
| | XSleepNet [10] | 0.783 | 0.761 | 0.725 | 0.708 | 0.689 | 0.615 |
| | CIMSleepNet | **0.801** | **0.794** | **0.751** | **0.788** | **0.777** | **0.726** |
| MHR | FeatConcat | 0.700 | 0.464 | 0.237 | 0.477 | 0.243 | 0.011 |
| | MLP [24] | 0.723 | 0.529 | 0.306 | 0.610 | 0.348 | 0.035 |
| | DeepCNN [9] | **0.759** | **0.615** | **0.421** | 0.616 | 0.354 | 0.039 |
| | CIMSleepNet | 0.729 | 0.553 | 0.348 | **0.701** | **0.466** | **0.240** |

and SHHS; EEG, EOG and EMG, for SVUH-UCD; motion signal and HR, for MHR. We provide detailed introduction and preprocessing methods of all datasets in Appendix E.

**Implementation Details**: In the first four datasets, CIMSleepNet is trained and tested using $k$-fold cross-validation, with a total of five repetitions of this procedure. Each result is the average of five results. In the last dataset, the training strategy refers to [26]. The detailed experimental settings and important hyperparameter settings are in Appendix F.

## 4.2 Comparison with the state-of-the-arts

In **randomly partially missing** case, we compare our CIMSleepNet with 8 ASS methods that can support multimodal learning: FeatConcat, MultitaskCNN [8], SalientSleepNet [23], MM-Net [11], TransSleep [16], XSleepNet [10], MLP [24] and Deep-CNN [9]. We leverage mask matrix $\mathbf{Z}$ and the public code of these methods to simulate the incomplete modality case. Then, we compare CIMSleep-Net with them under different missing rate $\rho$. According to the calculation

Table 2: Performance comparison in completely missing case.

| Test Modalities | Methods | Acc | MF1 | $K$ |
|---|---|---|---|---|
| EEG | CoRe-Sleep [26] | 0.882 | 0.808 | 0.834 |
| | CIMSleepNet | **0.891** | **0.817** | **0.845** |
| EOG | CoRe-Sleep [26] | 0.853 | 0.753 | 0.792 |
| | CIMSleepNet | **0.858** | **0.760** | **0.798** |
| EEG+EOG | CoRe-Sleep [26] | 0.895 | 0.823 | 0.853 |
| | CIMSleepNet | **0.903** | **0.828** | **0.862** |

formula of $\rho$, for two modalities, the missing rate ranges from [0.0, 0.1, 0.2, 0.3, 0.4, 0.5]; for three modalities, the missing rate ranges from [0.0, 0.1, 0.2, 0.3, 0.4, 0.5, 0.6, 0.7], where 0.7 is an approximate value of 2/3. In the **completely missing case**, we compare CIMSleepNet with CoRe-Sleep [26], the only existing ASS method that can handle complete missing of one or more modalities. We employ accuracy (Acc), macro F1-score (MF1) and Cohen Kappa ($K$) [44] to quantitatively analyze all methods. We also compare the data recovery performance of SMCCL with ICL [28] and SCL [30]. All methods are described in Appendix G.

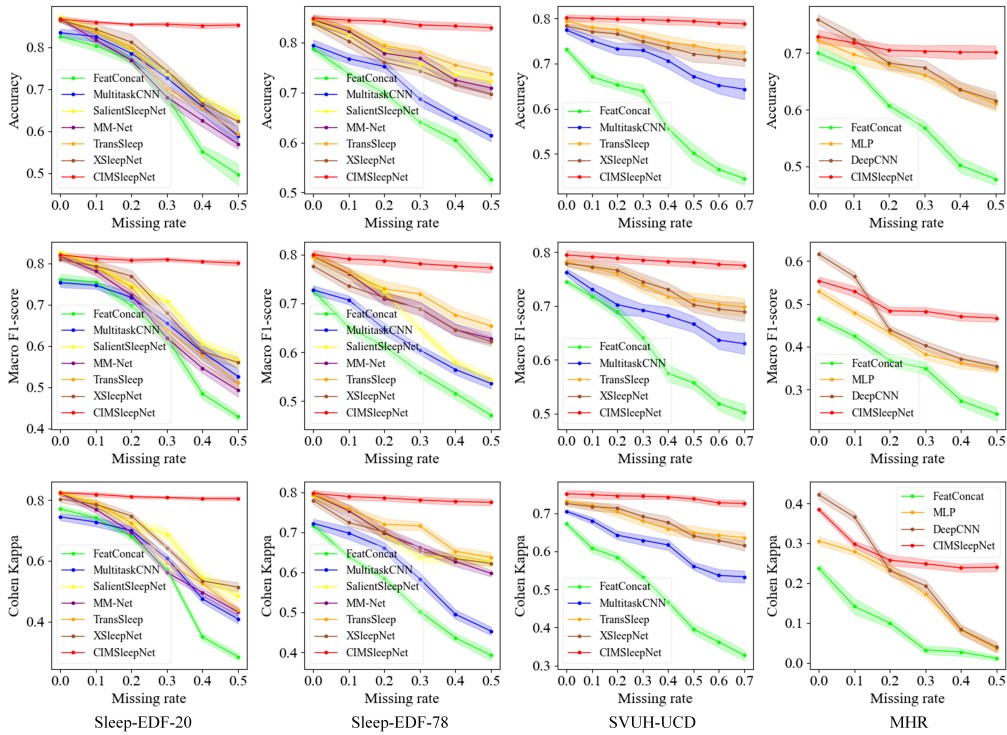

Figure 4: Impact of various missing rates.The shaded area represents the range of upper and lower standard deviations.

**Quantitative results**: As shown in Tab 1, CIMSleepNet achieves performance comparable to the state-of-the-arts in the complete modality. In the incomplete modalities, compared to the performance on complete modalities, all models exhibit a decrease in performance on the four datasets. Fortunately, CIMSleepNet has the least performance degradation and performs the best. As schematized in Fig. 4, we further evaluate the performance of CIMSleepNet and other methods under different missing rates. We observe that CIMSleepNet outperforms other methods in almost all datasets and missing rates. As the missing rate increases, the performance of other methods begins to decline significantly. Relatively speaking, CIMSleepNet exhibits a more stable trend. Further, Tab 2 exhibits the performance of CIMSleepNet trained with $\rho = 0.5$ (maximizing the model's robustness to missing modalities) and tested under complete modality absence. We observe that CIMSleepNet outperforms CoRe-Sleep in terms of performance across different testing modalities.

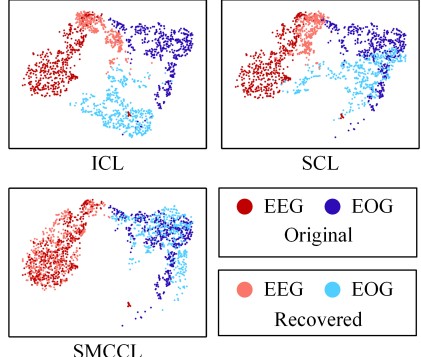

Figure 5: Visualization of the recovered modalities by ICL, SCL and SMCCL.

**Qualitative results**: We substitute ICL and SCL with SMCCL on CIMSleepNet to compare the performance of these three contrastive learning methods in data recovery (when $\rho = 0.5$). As depicted in Fig. 5, we randomly selected 500 recovered missing samples (500 EEG epochs and 500 EOG epochs) in Sleep-EDF-20 and projected them into 2D space via t-SNE [45]. ICL only focuses on the inter-modal consistency and ignores the recovery of semantic information. SCL retains semantic information based on ICL, thereby improving data matching. However, ICL and SCL tend to learn the inter-modal consistency, i.e., utilize multimodal shared information to guide the recovery of missing data. This strategy easily leads to the loss of modality-specific information, thus failing to exploit the inter-modal complementarity. Different from ICL and SCL, our SMCCL explores the intrinsic connection between semantic and modal information under mutual information theory. Hence, compared to ICL and SCL, the data recovered by SMCCL exhibits a more consistent distribution

with the original data, further demonstrating its effectiveness in handling missing modalities. We also visualize the features extracted by each method. Specifically, we randomly select the data of one subject (9th) among the Sleep-EDF-20. As illustrated in Fig. 6, we use t-SNE [45] to visualize the distribution of features generated by all methods at $\rho = 0.5$, which are extracted before the final decision head. Compared with other methods, our CIMSleepNet can extract more discriminative representations in incomplete modalities, further demonstrating its robustness.

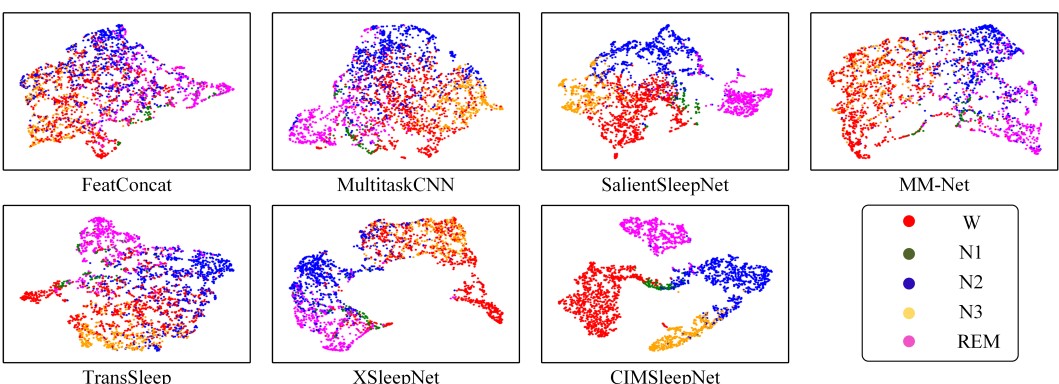

Figure 6: Visualization of latent features of different methods on Sleep-EDF-20.

**Ablation studies**: We conduct ablation studies for CIMSleepNet on Sleep-EDF-20 under the condition of missing rate $\rho = 0.5$. It can be observed from Tab 3 that no matter which component is deleted, each evaluation metric of the results will decrease. It is particularly noteworthy that in the absence of both MAIM and SMCCL, the performance drops significantly, further demonstrating their importance in dealing with the missing modality issue. Furthermore, we find that although the two components designed to mitigate modality missing issue (MAIM and SMCCL) introduce additional parameters, the increase is much less than that introduced by the sequence modeling component (MCTA). However, sequence modeling is crucial for capturing the temporal information of PSs and improving model performance [10, 46]. The ablation experiment of MCTA, parameter sensitivity analysis and training process analysis are detailed in Appendix H, I and J, respectively.

Table 3: Ablation study of CIMSleepNet on Sleep-EDF-20. "✓" indicates the use of this component. MCTA indicates the Transformer equipped with MCTA. The context length of single inference is 25.

| MAIM | SMCCL | MCTA | Acc | MF1 | $K$ | Model Size (MB) | GFLOPs |
|------|-------|------|-----|-----|-----|-----------------|--------|
|      |       |      | 0.497 | 0.429 | 0.285 | 2.344 | 0.069 |
| ✓    |       |      | 0.771 | 0.704 | 0.672 | 5.767 | 0.096 |
|      | ✓     |      | 0.786 | 0.726 | 0.699 | 8.458 | 0.071 |
|      |       | ✓    | 0.694 | 0.629 | 0.536 | 30.272 | 2.206 |
| ✓    | ✓     |      | 0.810 | 0.756 | 0.759 | 4.412 | 0.097 |
| ✓    |       | ✓    | 0.829 | 0.778 | 0.777 | 33.696 | 2.876 |
|      | ✓     | ✓    | 0.834 | 0.786 | 0.784 | 36.386 | 2.246 |
| ✓    | ✓     | ✓    | **0.853** | **0.801** | **0.805** | 37.678 | 2.902 |

## 5 Conclusion

We try to challenge multimodal ASS under incomplete modalities by proposing CIMSleepNet. In CIMSleepNet, MAIM reconstructs missing modality data by establishing interactions among modalities, which allows for the provision of complete modality data support for subsequent components. Meanwhile, SMCCL ingeniously leverages semantic information and modal information to subdivide similarity into three levels, thereby simulating real data distribution. Then, MCTA mechanism accomplishes comprehensive temporal context modeling, further improving the expressive ability of latent temporal representations. Extensive experiments demonstrate that the effectiveness of CIMSleepNet in various incomplete modalities.

## Acknowledgments

This work was supported by the National Natural Science Foundation of China (62072089) and the Fundamental Research Funds for the Central Universities of China (N2116016, N2224001-10).

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

## A    Limitations

Our work also has some limitations. To deal with the missing modality issue and temporal dependency, we introduce an additional architecture, which will incur additional computational overhead. It is worth mentioning that the lack of labeling information is also a common phenomenon in real-world applications. Hence, we expect to develop an unsupervised or semi-supervised learning multimodal approach in our forthcoming study, which can simultaneously address the challenges of modality missing and label missing.

## B    Broader Impacts

Our work facilitates the daily sleep quality assessment and sleep disorder diagnosis for the public, and lays the foundation for promoting personalized treatment of sleep disorders. However, the multimodal physiological data required for model training may involve sensitive personal health data, which may bring potential social privacy and security issues.

## C    Similarity Weight Matrix Construction Example

Supposing that a batch contains 3 multimodal epochs and the number of modal types is 3, the calculation process and results of the similarity weight matrix $\mathbf{W}$ are illustrated in Fig. 7. Specifically, the redefinition of matrices $\mathbf{Y}$ and $\mathbf{S}$ is realized by using flatten and replicate operations. Moreover, calculate $\bar{\mathbf{Y}}$ and $\bar{\mathbf{S}}$ separately to obtain the mask matrices $\mathbf{U}$ and $\mathbf{V}$ by

$$\mathbf{U} = \{\{u_i^j\}_{i=1}^R\}_{j=1}^R, u_i^j = \left\{ \begin{array}{ll} 1, & \bar{y}_i^j = \dot{y}_i^j \\ 0, & \bar{y}_i^j \neq \dot{y}_i^j \end{array} \right. \quad \mathbf{V} = \{\{v_i^j\}_{i=1}^R\}_{j=1}^R, v_i^j = \left\{ \begin{array}{ll} 1, & \bar{s}_i^j = \dot{s}_i^j \\ 0, & \bar{s}_i^j \neq \dot{s}_i^j \end{array} \right. \tag{11}$$

where "1" is a positive pair and "0" is a negative pair. Besides, $\dot{y}_i^j$ and $\dot{s}_i^j$ are the elements in $\bar{\mathbf{Y}}^{\mathrm{T}}$ and $\bar{\mathbf{S}}^{\mathrm{T}}$ respectively. Further, combining modality consistency matrix $\mathbf{\Theta}$, the similarity weight matrix $\mathbf{W}$ can be constructed by

$$\mathbf{W} = \underbrace{\mathbf{U} \odot \mathbf{V}}_{\text{the 1th level}} + \underbrace{(1\text{-}\mathbf{\Theta})(\mathbf{U} - \mathbf{U} \odot \mathbf{V})}_{\text{the 2th level}} + \underbrace{\mathbf{\Theta}(\mathbf{V} - \mathbf{U} \odot \mathbf{V})}_{\text{the 3th level}} \tag{12}$$

where $\odot$ denotes element-wise multiplication and $\mathbf{\Theta}$ in Fig. 7 is composed of the set consisting of $\theta_1$, $\theta_2$ and $\theta_3$.

## D    Theoretical Proof

We simplify $\frac{I(\phi^k; \phi^i)}{H(\phi^k, \phi^i)}$ in continuous random variables. It can be expressed as follows

$$\begin{aligned}
\frac{I(\phi^k; \phi^i)}{H(\phi^k, \phi^i)} &= \frac{H(\phi^k) + H(\phi^i) - H(\phi^k, \phi^i)}{H(\phi^k, \phi^i)} \\
&= \frac{\int p(x)\ln\frac{1}{p(x)}dx + \int p(y)\ln\frac{1}{p(y)}dy - \iint p(x,y)\ln\frac{1}{p(x,y)}dxdy}{\iint p(x,y)\ln\frac{1}{p(x,y)}dxdy} \\
&= \frac{\iint p(x,y)\ln\frac{1}{p(x)}dxdy + \iint p(x,y)\ln\frac{1}{p(y)}dxdy - \iint p(x,y)\ln\frac{1}{p(x,y)}dxdy}{\iint p(x,y)\ln\frac{1}{p(x,y)}dxdy} \\
&= \frac{\iint p(x,y)\ln\frac{1}{p(x)p(y)}dxdy - \iint p(x,y)\ln\frac{1}{p(x,y)}dxdy}{\iint p(x,y)\ln\frac{1}{p(x,y)}dxdy} \\
&= \frac{\iint p(x,y)\ln\frac{p(x,y)}{p(x)p(y)}dxdy}{\iint p(x,y)\ln\frac{1}{p(x,y)}dxdy} \\
&= \iint \log_{\frac{1}{p(x,y)}}\left(\frac{p(x,y)}{p(x)p(y)}\right)dxdy
\end{aligned} \tag{13}$$

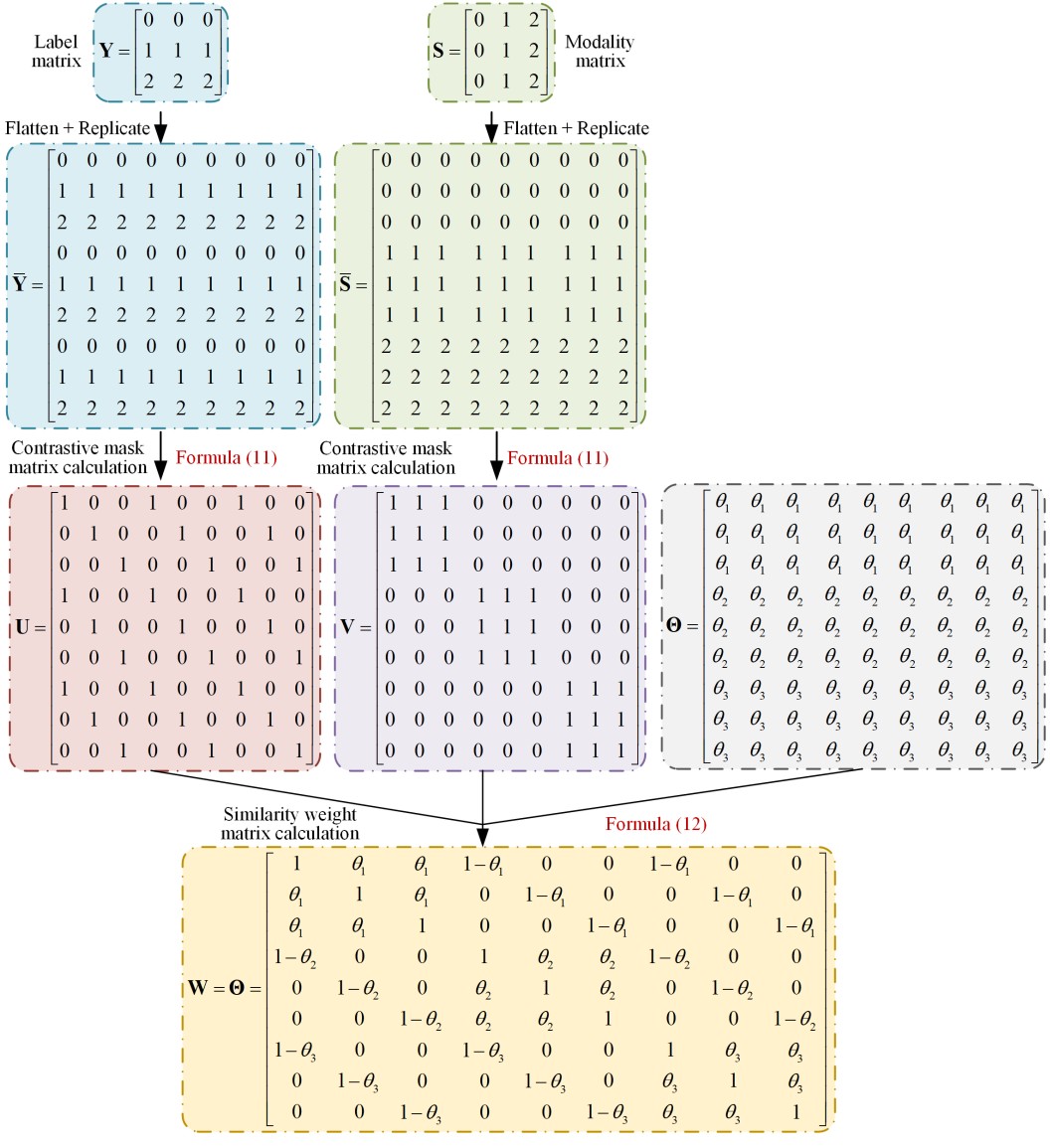

Figure 7: Example of similarity weight matrix $\mathbf{W}$ construction.

where $p(x)$ and $p(y)$ are the marginal probability distributions of $\phi^k$ and $\phi^i$, respectively, i.e., the continuous form of $\mathcal{P}(m)$ and $\mathcal{P}(n)$ in formula (6). $p(x,y)$ is their joint probability distribution, i.e., the continuous form of $\mathcal{P}(m, n)$ in formula (6). Therefore, the expression of formula (6) is obtained.

# E    Data and Preprocessing

1) **Sleep-EDF-20** [40, 41]: The dataset has been widely applied in sleep study, comprising 39 nights of PSG recordings from 20 subjects. The subjects are aged between 25 and 34 years old, with 10 males and 10 females. Each recording are divided into epochs in units of 30 seconds. The data preprocessing method draws on the previous work [10]. After preprocessing, the context length $T$ is set to 25, and the redundant segments at the front end are discarded.. These epochs are classified into five different categories, including wake (W), rapid eye movement (REM), and three types of non-REM (N1, N2, and N3). Then, two modalities, electroencephalogram (EEG) (Fpz-Cz channel) and electrooculogram (EOG) (ROC-LOC channel), are utilized to evaluate CIMSleepNet. Among them, the sampling frequency of EEG and EOG is both 100Hz.

2) **Sleep-EDF-78** [40, 41]: The dataset includes 153 recordings from 78 subjects. The subjects have a wide age range, from 25 to 101 years old, and included 41 males and 37 females. Similarly, the length of each epoch is 30 seconds, and the method [10] is utilized to preprocess Sleep-EDF-78 data. We set the context length $T$ to 25 for each modality. The choice of modalities, categories and sampling frequency are the same as for Sleep-EDF-20.

3) **SVUH-UCD** [40]: The dataset focuses on sleep staging study with sleep disorders. It includes 25 PSG recordings from 25 sleep apnea patients. Their ages ranged from 28 to 68, including 21 males and 4 females. Following previous study [47], we choose EEG (C3-A2 channel, 128 Hz), EOG (horizontal channel, 64 Hz) and EMG (64 Hz), and resample these recordings to 100Hz. Furthermore, we also set the context length $T$ to 25. This dataset are also divided into five sleep stages.

4) **MHR** [24]: This is a public sleep dataset based on wearable devices, which contains overnight sleep recordings from 31 subjects. Subjects are allowed 8 hours of sleep monitoring opportunities and each epoch is 30 seconds in length. We preprocess this dataset using the method described in previous work [24]. After the preprocessing is completed, we set the context length $T$ to 20 and exclude redundant epochs. Following the usage rules of this dataset, we employ two modalities, the motion signals composed of three-axis accelerometry and the heart rate signals, to perform classification tasks for the three categories: W, non-REM, and REM. In these two modalities, the sampling frequency of motion signal and heart rate signal is 50Hz and 1Hz respectively.

5) **SHHS** [42, 43]: This dataset is a large sleep dataset collected from multiple sleep centers, which contains two sub-datasets, namely SHHS-1 and SHHS-2. Following previous studies [10, 26], we choose SHHS-1 for our experiment. SHHS-1 consists of 5,791 subjects aged between 39 and 90 years old. We employ the EEG (C4-A1 channel, 125Hz) and EOG (L-R channel, 50Hz), and resample them to 100 Hz. Furthermore, the context length $T$ is set to 25. We also divide it into five sleep stages.

## F Implementation Details

We choose the programming language based on Python 3.8 and the deep learning framework based on PyTorch 1.13 to build and train the model. All experiments are conducted on a server containing an RTX 4090 GPU (24GB) and an Intel(R) Xeon(R) Gold 6430 processor (120GB) equipped with 16 virtual CPUs. The total objective loss is mainly optimized through the Adam optimizer. In the first four dataset, CMISleepNet is trained (Held-out validation set 4 subjects for Sleep-EDF-20, SVUH-UCD and MHR; 7 subjects for Sleep-EDF-78) and tested by $k$-fold cross-validation. Specifically, CMISleepNet performs five random samplings and five $k$-fold cross-validations for each missing rate in every dataset. After each $k$-fold cross-validation, the prediction results from the test sets of all folds are combined as one time result. Each missing rate result is the average of five results. In the last dataset, the training strategy refers to [26], i.e., using a random split of 0.7 and 0.3 for the train (Held-out 100 subjects for validation) and test set.

The important hyperparameters on different datasets can be described as: We set the learning rate of all datasets to 0.001 and 0.0001 before and after the 10th iteration, respectively. The weight decay is set to 0.0001 for all datasets. The maximum number of iterations is set to 100 for all datasets. The number of intra-epoch heads $S_1 = 4$ and the number of inter-epoch heads $S_2 = 8$ for all datasets. The number of cross-validation folds, $k$, is set to 20 for Sleep-EDF-20; 20 for Sleep-EDF-78; 25 for SVUH-UCD and 15 for MHR. The coefficient set $\tilde{\mathbf{W}}$ utilized to adjust category weights are set to [1.5, 2.5, 1.5,1.0, 1.5] for Sleep-EDF-20; [1.5, 2.2, 1.5,1.0, 1.5] for Sleep-EDF-78; [1.5, 2.0, 1.5,1.0, 1.5] for SVUH-UCD; [ 2.0, 1.0, 2.0] for MHR; [2.0, 3.0, 1.5,1.0, 1.5] for SHHS. Further, we set the coefficients $\alpha$ and $\beta$ of the total objective function to 0.001 and 0.01, respectively for all datasets.

## G Compared Methods

In **randomly partially missing** case, we compare our CIMSleepNet with 8 ASS methods that can support multimodal learning: FeatConcat, MultitaskCNN [8], SalientSleepNet [23], MM-Net [11], TransSleep [16], XSleepNet [10], MLP [24] and DeepCNN [9]. In the **completely missing case**, we compare CIMSleepNet with CoRe-Sleep [26], the only existing ASS method that can handle complete missing of one or more modalities. We also compare the data recovery performance of SMCCL with ICL [28] and SCL [30]. All methods are described as follows.

1) **FeatConcat**: The temporal CNNs are utilized as the feature extractor of each modality data, and the features of different modalities are directly fused. Moreover, the fused features are sent to MLP for classification.

2) **MultitaskCNN** [8]: Based on the sleep staging task, the task of predicting adjacent epochs is added to improve the multimodal sleep staging performance.

3) **SalientSleepNet** [23]: A dual-branch U2-Net structure is proposed to improve the feature extraction of salient waves of multimodal physiological signals.

4) **MM-Net** [11]: A cross-link fusion module is exploited to reduce redundant information of multi-modality and multi-view.

5) **TransSleep** [16]: Two auxiliary tasks, segment confusion stage estimation and stage-transition detection, are designed to address stage transitions during sleep. We make it capable of handling multimodal data by increasing the number of channels at the input head of TransSleep.

6) **XSleepNet** [10]: An adaptive gradient blending strategy is designed to improve the joint representation ability of the original signal and the corresponding time-frequency image.

7) **MLP** [24]: Combining motion features and heart rate features, and using MLP to implement sleep staging (Wake/ NREM/ REM) based on wearable devices.

8) **DeepCNN** [9]: A deep CNN is constructed to explore the impact of early-stage fusion, late-stage fusion and hybrid fusion. We chose the late-stage fusion solution because it has the best performance.

9) **CoRe-Sleep** [26]: The ingenious combination of self-attention and cross-attention improves the robustness of the model under imperfect data.

10) **ICL** [28]: Improving modal consistency by bringing different modalities of the same instance closer together.

11) **SCL** [30]: The introduction of semantic information improves the ability to recover the semantic structure information of data.

## H   Ablation studies of MCTA

We also explore the internal details of Transformer equipped with MCTA. The six baseline models in Tab 4 are substructures of Transformer equipped with MCTA. Among them, Intra-*X* denotes using two *X* layers for intra-epoch temporal dependency modeling. Inter-*X* denotes using two *X* layers for inter-epoch context modeling. Intra & Inter-*X* denotes using four *X* layers for temporal context modeling, with the first two layers capturing intra-epoch temporal dependency and the last two layers capturing inter-epoch context. *X* refers to GRU or Transformer. It can be observed from the results that both Intra & Inter-GRU and Intra & Inter-Transformer outperform their respective single-level models. Further, CIMSleepNet performs the best when using Transformer equipped with MCTA, which proves the effectiveness of multi-level cross-branch representation fusion.

Table 4: Ablation study of Transformer equipped with MCTA on Sleep-EDF-20.

| Methods | Acc | MF1 | *K* |
|---|---|---|---|
| Intra-GRU | 0.827 | 0.775 | 0.772 |
| Inter-GRU | 0.835 | 0.780 | 0.787 |
| Intra & Inter-GRU | 0.839 | 0.788 | 0.791 |
| Intra-Transformer | 0.813 | 0.770 | 0.765 |
| Inter-Transformer | 0.837 | 0.789 | 0.793 |
| Intra & Inter-Transformer | 0.845 | 0.795 | 0.797 |
| Transformer with MCTA | **0.853** | **0.801** | **0.805** |

# I  Parameter Sensitivity Analysis

We explore the impact of $\alpha$ and $\beta$ on the performance of CIMSleepNet on the Sleep-EDF-20. As shown in Fig. 8, as the values of $\alpha$ and $\beta$ change, the performance of CIMSleepNet fluctuates to varying degrees. We also observe that $\beta$ has a greater sensitivity to CIMSleepNet compared to $\alpha$.

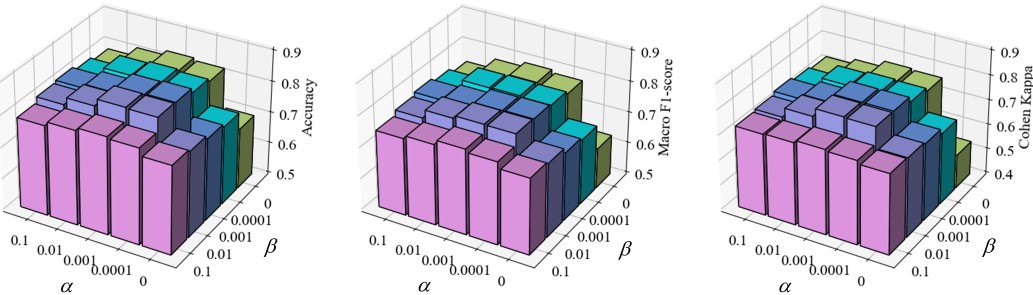

Figure 8: Hyperparameters, $\alpha$ and $\beta$, analysis on Sleep-EDF-20.

# J  Training Process Analysis

As schematized in Fig. 9, we provide visualizations of different validation loss curves to explore their real-time changes during training. We can observed: During the entire training process, modal imagination loss and distribution alignment loss will decrease as the number of iterations increases, which shows that the data imputation ability and distribution fitting ability of the model are gradually improving. It is worth mentioning that, in the early stages of training, the rate of decrease in distribution alignment loss is greater than that of modal imagination loss. This phenomenon occurs because the data generated during the initial training stage deviates significantly from the real distribution, requiring substantial adjustments through distribution alignment loss function. When the modal imagination loss and distribution alignment loss are in a stationary state, the classification loss continues to decrease, which indicates that the data generated in the stationary state will further improve the classification performance of the model.

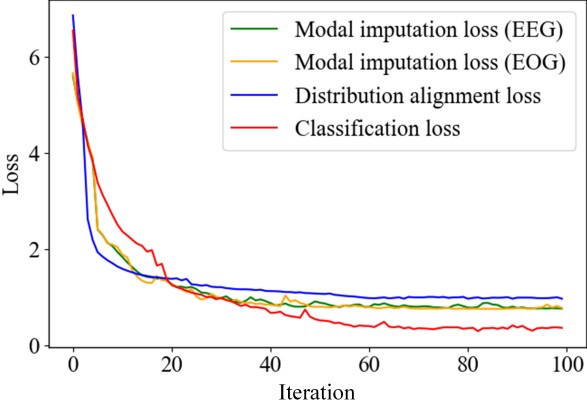

Figure 9: Training dynamics of modal imagination loss, distribution alignment loss and classification loss on Sleep-EDF-20. Among them, modal imagination loss is presented in two modalities: EEG and EOG respectively.

