# OpenReview forum: "Robust Sleep Staging over Incomplete Multimodal Physiological Signals via Contrastive Imagination"
_NeurIPS.cc/2024/Conference — NeurIPS 2024 poster_

### Official Review · Reviewer_HATA · 2024-07-08

**Soundness:** 3
**Presentation:** 3
**Contribution:** 3
**Rating:** 6
**Confidence:** 4

**Summary:**

This manuscript proposes an end-to-end framework that incorporates many procedures into an automatic sleep staging system that is specifically designed in the context of missing data.  This includes a module for missing modality imputation, a module for distribution alignment, and a module for feature extraction and temporal context modeling.  The net result of these modifications is that the approach appears to work much better than competing algorithms in the context of missingness.

**Strengths:**

This manuscript is fairly clear on its overall concepts.  The scientific motivation is well-justified, as the missing data is a problem in practice and has received significantly less attention in the field.  There are several novel contributions in the different stages of modeling.

**Weaknesses:**

There are several minor weaknesses.

First, it is unclear how the different comparison methods are used in the presence of missing data.  It sounds like the missing data is just removed from the data and sent through the comparison methods without modification.  However, that would be a departure from typical practice and represent an unrealistic scenario.  Note that EEG is routinely screened and bad channels are replaced by statistical inference (such as in HAPPE, MNE, FIELDTRIP, etc.).  As such, the fair comparison would be to a 2 stage procedure where the data is first imputed and then put through the framework.   In my experience, you see very little impact on performance for tasks like this up to a reasonable amount of missingness, certainly much less than the impact reported in Figure 4.

While the proposed techniques are mathematically nuanced, it is not always clear why they work.  For example, in the qualitative results, SMCCL is matching better than ICL or SCL, but it is not clear why that is happening to me.  Given that the data is recovered, I would have expected the distributions to match better.  I would appreciate a bigger commentary here.  I have the same issue with MTCA in the appendix, where there is a comparison table but not a clear description of why the performance differences are happening.

I found the discussion of existing work on multi-modality with missing data to be a bit sparse.  This is important context for this work, as there is a fair amount of work on missingness in multi-modality previously in different scientific domains and it is not compared to here.  At a minimum, I would appreciate an increased discussion of these methods and a discussion of why they were not appropriate as comparison methods.

**Questions:**

As above, how does a more standard two-step procedure (infer missingness, then apply algorithm) perform compared to these current methods?

Why do ICL and SCL not infer the same distribution on the recovered samples (Figure 5), as the recovered samples should largely match the data distribution?

Why are the uncertainties in Figure 4 so small?  For example, in Sleep-EDF-20, there are only 20 subjects and I would have expected a larger standard deviation.

**Limitations:**

See first question above.

---

> ### Author Rebuttal · Authors · 2024-08-07
>
> **Weaknesses:**
>
> R1: Thank you for your valuable comments. Similar to mainstream sleep staging methods [1][2], the EEG, EOG, and EMG signals we use are all single-channel. Taking EEG as an example, multi-channel EEG requires expensive equipment such as PSG with strict environmental requirements for collection. In contrast, single-channel EEG is easier to collect with portable devices and reduces non-invasiveness to subjects. Single-channel signals are not limited to a specific collection environment and are more conducive to the promotion of sleep stages. This means that in actual applications, we will not collect multi-channel data. Hence, the method of using other channels to perform statistical inference on the missing channel is not applicable to our study. In addition, our experimental scheme for the missing modality is consistent with previous work [2], which are all single-step.
>
> However, interpolation methods in the temporal dimension are applicable. As depicted in Figure 1 from the PDF, we explore the performance of five models at various missing rates. Among them, BaseSleepNet is the baseline model after CIMSleepNet removes MAIM and SMCCL. We perform nearest neighbor interpolation (NNI), 3rd order polynomial interpolation (3PI) and cubic spline interpolation (CSI) on the subjects' nightly recordings. Then, the three interpolated data are fed into BaseSleepNet. We observe that the three interpolation methods only slightly improve the performance of BaseSleepNet.  Especially as the missing rate increases, they can even negatively impact BaseSleepNet's performance. Temporal imputation of missing values can lead to semantic information confusion, thus interfering with sleep staging. CIMSleepNet achieves an integration of 'missing imputation-sleep staging'. The end-to-end method reduces the complexity and potential errors associated with multi-step processing, avoiding inconsistencies between different steps.
>
> R2: Thank you for your valuable suggestions. Since the multimodal data in the training set is also incomplete, MAIM learns from the available modalities in the incomplete multimodal data and utilizes other available data within the same modality for guidance to recover the missing data. This is an appropriate strategy, but it can lead to modality and semantic biases [3]. As shown in Figure 5 from the manuscript, ICL only focuses on the inter-modal consistency and ignores the recovery of semantic information. SCL retains semantic information based on ICL, thereby improving data matching. However, ICL and SCL tend to learn the inter-modal consistency, i.e., utilize multimodal shared information to guide the recovery of missing data. This strategy easily leads to the loss of modality-specific information, thus failing to exploit the inter-modal complementarity. Different from ICL and SCL, our SMCCL explores the intrinsic connection between semantic and modal information under mutual information theory. The designed three-level similarity adaptively introduces semantic and modal information into the restored data, so that it not only retains the semantic structure but also restores the specific modal information to a large extent. Hence, as indicated in Figure 5, the data restored by SMCCL is more inclined to the real data distribution.
>
> There are two reasons why MCTA achieves optimal performance. Firstly, MCTA not only realizes the mining of intra-epoch (local) temporal information, but also realizes inter-epoch (global) contextual learning. Compared with single-level methods, MCTA shows a more comprehensive sequence modeling ability. Secondly, MCTA realizes the interaction between self-attention weights and recurrent bias, which not only enables efficient context learning, but also improves the shortcomings of the Transformer's insufficient recurrent modeling ability.
>
> R3: We discuss five methods related to our study in the related work section. One is the only existing incomplete multimodal sleep staging method. The other four methods use ICL or SCL to solve the missing modality, which is related to our SMCCL. Although we cannot directly apply these four methods to sleep staging, ICL and SCL can be embedded in our method. This is why we start with contrastive learning to introduce the work related to missing modality. We will further add other related work on missing modality to clarify why they cannot be directly applied to sleep staging. Moreover, they require complete training sets, which contradicts the practical applications of sleep staging.
>
> **Questions:**
>
> R1: Please refer to R1 in **Weaknesses**.
>
> R2: Please refer to R2 in **Weaknesses**.
>
> R3: We use $k$-fold cross-validation for each training, which allows us to fully utilize the entire dataset. Moreover, it reduces bias that may arise from a small data size and single dataset partitioning. Take Sleep-EDF-20 as an instance, it contains 39 nights of recordings from 20 subjects. Each sample is 30 s, so the sample size is relatively sufficient to ensure the stable performance in each training. For sleep-EDF-20, all subjects are healthy and within a small age range (25 and 34 years old). Hence, on Sleep-EDF-20, the small individual differences are also an important reason for the smallest uncertainty and the best performance. Thanks to the support of MAIM and SMCCL, the high robustness of CIMSleepNet is also one of the reasons for its small performance uncertainty.
>
>
>
>
>
>
>
>
> [1] Phan H, Chén OY, Tran MC, et. al. XSleepNet: Multi-view sequential model for automatic sleep staging. IEEE Transactions on Pattern Analysis and Machine Intelligence. 2021, 4(9): 5903-5915.
>
> [2] Kontras K, Chatzichristos C, Phan H, et. al. CoRe-Sleep: A Multimodal Fusion Framework for Time Series Robust to Imperfect Modalities. IEEE Transactions on Neural Systems and Rehabilitation Engineering. 2024, 32: 840-849.
>
> [3] Qian S, Wang C. COM: Contrastive Masked-attention model for incomplete multimodal learning. Neural Networks. 2023, 162: 443-455.

---

> > ### Comment · Reviewer_HATA · 2024-08-09
> > **Further question on uncertainty estimation**
> >
> > Can you please provide the mathematical definition for how you are estimating uncertainty?  e.g., are you calculating it over subject or over samples?   Generally the increase of more samples from a single subject doesn't decrease uncertainty by very much, since the performance of interest is over subject.
> >
> > Second, can you confirm that the K-fold cross-validation is over subjects and not samples?

---

> > > ### Author Response · Authors · 2024-08-09
> > > **Detailed response regarding uncertainty results**
> > >
> > > Thank you very much for giving us your valuable comments during your busy schedule. To evaluate the performance of the model under different random missing cases, we set five different random missing seeds for each missing rate and perform five training sessions respectively. Then, we take the average and standard deviation of these five evaluation metrics. We mainly use the upper and lower standard deviations of these five results to reflect the uncertainty of the model.
> > >
> > > For the five training sessions mentioned above, similar to previous work on sleep staging [1][2], we performed $k$-fold cross-validation for each training session. Taking into account the influence of individual differences, the $k$-fold cross-validation we used is an over subjects, i.e., subject-independent strategy. Suppose there are $N$ subjects in a dataset. Each fold will utilize $N-M$ subjects for training and the remaining $M$ subjects for testing.  $\text{If } i < k$, $M=N - \left\lceil \frac{N}{k} \right\rceil$, and $ \text{if } i = k$, $M=N - (k - 1) \left\lceil \frac{N}{k} \right\rceil$, where $\left\lceil {} \right\rceil $ indicates round up. After $k$-fold cross-validation, the predicted results of the test sets in all folds are combined for overall evaluation like [1][2]. Therefore, each evaluation metric is predicted on the entire dataset, which may be one of the reasons for the small uncertainty. Moreover, the standard deviation mentioned above is not calculated based on the test results of each fold, but is obtained based on five random missing cases.
> > >
> > >
> > >
> > >
> > >
> > >
> > >
> > >
> > >
> > >
> > > [1] Phan H, Chén OY, Tran MC, et. al. XSleepNet: Multi-view sequential model for automatic sleep staging. IEEE Transactions on Pattern Analysis and Machine Intelligence. 2021, 4(9): 5903-5915.
> > >
> > > [2] Phyo J, Ko W, Jeon E, Suk HI. TransSleep: Transitioning-aware attention-based deep neural network for sleep staging. IEEE Transactions on Cybernetics. 2022, 53(7): 4500-4510.

---

> ### Author Response · Authors · 2024-08-14
>
> Dear reviewer HATA,
>
> We appreciate your careful review and valuable comments on the manuscript. We attach great importance to your feedback and have started to revise and improve it. In addition, we have clarified some of your concerns.
>
> We understand that the discussion period is about to end, so we will remain online in the next period of time. If you have any questions or suggestions, we are very willing to communicate further and try our best to answer any concerns you have about the manuscript.
>
> Thank you again for your support and help in our work.
>
> Sincerely,
>
> Authors

---

### Official Review · Reviewer_QG76 · 2024-07-10

**Soundness:** 3
**Presentation:** 3
**Contribution:** 3
**Rating:** 7
**Confidence:** 3

**Summary:**

This paper presents CIMSleepNet, a framework designed to address the challenges of automated sleep staging when faced with incomplete multimodal physiological data. This framework is particularly relevant in real-world applications where sensor malfunctions or detachment often results in incomplete data, thereby affecting the performance of sleep staging systems.

**Strengths:**

Thanks for your work! Here are my comments:
1. CIMSleepNet effectively addresses the common issue of missing modalities in sleep stage classification through its Modal Awareness Imagination Module (MAIM) and the proposed Semantic & Modal Calibration Contrastive Learning (SMCCL). This innovative combination helps in approximating the missing modal data and aligning it closer to real data distributions better while comparing to other methods.
2. The framework is validated across five different multimodal sleep datasets, showing its robustness and effectiveness in improving sleep staging performance.
3. The multi-level cross-branch temporal attention mechanism allows the framework to capture comprehensive temporal context information, which can enhance the model's ability to interpret complex stage-transitioning patterns using PSs.

**Weaknesses:**

1. The architecture integrates multiple sophisticated components, which could potentially increase the computational overhead, making it less efficient for real-time applications, also, the training cost of time for this model is not given out.

**Questions:**

1. Scalability Issue: It would be interesting to see how does CIMSleepNet scale with even larger datasets or more varied modal distributions? Is its performance consistent across other non-sleep related tasks?
2. Potential Extension: Can the method be extended to handle missing labels in addition to missing modalities, addressing semi-supervised learning scenarios?
3. Time Cost Issue: What is the impact on inference time when using the proposed framework?

**Limitations:**

1. Computational Cost: Increased computational overhead from additional architecture components.
2. Modality and Data Limitation: Tested on a limited set of modalities and datasets.

---

> ### Author Rebuttal · Authors · 2024-08-07
>
> **Weaknesses:**
>
> R1: Thank you for your valuable suggestions! We have further analyze the computational efficiency and resource requirements of CIMSleepNet. Please refer to the Author Rebuttal for details.
>
> **Questions:**
>
> R1: As schematized in Table 2 of the original manuscript, we have compared the currently only state-of-the-art incomplete multimodal sleep staging method on the largest publicly available multimodal sleep staging dataset, i.e., the SHHS dataset. CIMSleepNet exhibits significant advantages in both complete and incomplete multimodal  cases. Moreover, as reported in Table 3 from the PDF, we further demonstrate the superiority of CIMSleepNet over other state-of-the-art multimodal sleep staging methods. These advantages mainly stem from the excellent data recovery ability of MAIM combined with SMCCL and the powerful sequence modeling ability of MCTA. Current multimodal sleep staging methods mainly rely on EEG, EOG, and EMG signals, which are also the data used by most previous sleep staging studies and our study. Bseides, we apply our method to heart rate and motion signals collected from real wearable devices and verify the generalization ability of the model across different modalities. In fact, the multimodal dataset used in this study already covers the main physiological signals commonly used in current sleep staging study [1]. These signals have been widely verified and recognized and can be effectively used for sleep staging.
>
> As reported in Table 4 of the PDF, we conduct experiments on CIMSLeepNet using two non-sleep-related datasets, UCI-HAR [2] and WESAD [3]. The introduction and preprocessing of the two datasets are as follows:
>
> UCI-HAR: This is a public database applied to human activity recognition, which contains the daily activity recordings of 30 subjects. Subjects are instructed to wear a smartphone with embedded inertial sensors to perform signal acquisition for six physical activities. We set the context length $T$ to 20 and removed extra segments. The database includes three types of motion signals: three-axis total acceleration, three-axis body acceleration, and three-axis angular velocity. The sampling frequency of each motion signal is 50Hz, and the length of each segment is 2.56 s. We employ these three motion signals to conduct experiments on CIMSLeepNet.
>
> WESAD: The database, which focuses on study in the field of stress detection, includes physiological signals from 15 subjects. Many studies [4][5][6] have successively confirmed that  electrodermal activity (EDA)  andelectrocardiogram (ECG) have significant advantages in stress detection and emotion recognition. Therefore, we adopt these two modalities to achieve the validity verification of CIMSLeepNet. Refer to previous methods [5][6] to preprocess EDA and ECG. In the process, we also resampled the data from both modalities to 300Hz. In addition, the method of study [5] is adopted to divide the segments, which makes the length of each segment 10 s. After that, the context length $T$ to 20. Ultimately, CIMSLeepNet is utilized to perform discrimination between normal emotional and stressful states.
>
> During training, the number of cross-validation folds, $k$, is set to 10 for UCI-HAR and 15 for WESAD. The coefficient set $\smash{{{{\mathbf{\vec{W}}}}}}$ utilized to adjust category weights are set to [1.0, 1.0, 1.0, 1.0, 1.0, 1.0] for UCI-HAR and [1.0, 1.5] for WESAD.
>
> As reported in Table 4 from the PDF, our CIMSleepNet also performs well on two datasets in non-sleep related fields. Especially under incomplete modalities, CIMSleepNet still maintains good robustness. This further proves the effectiveness of CIMSleepNet in other tasks and multimodal signals.
>
> R2: We believe this is a very meaningful research direction. Since our method also encounters modality missing in the training set, it requires supervised information to guide both the modality recovery and sleep staging processes. Although our current work mainly focuses on handling missing modalities, in the future we will explore how to utilize partially labeled data and incomplete modalities to extend CIMSleepNet to semi-supervised learning scenarios.
>
> R3: Thank you for your valuable suggestions! We further analyze the computational efficiency, resource requirements, and inference time of CIMSleepNet. Please refer to the Author Rebuttal for details.
>
> **Limitations:**
>
> R1: Please refer to the **Author Rebuttal** for details.
>
> R2: Please refer to R1 in **Questions**.
>
> [1] Faust O, Razaghi H, Barika R, et. al. A review of automated sleep stage scoring based on physiological signals for the new millennia. Computer methods and programs in biomedicine. 2019, 176: 81-91.
>
> [2] Ignatov A. Real-time human activity recognition from accelerometer data using convolutional neural networks. Applied Soft Computing. 2018, 62: 915-922.
>
> [3] Schmidt P, Reiss A, Duerichen R, et. al. Introducing wesad, a multimodal dataset for wearable stress and affect detection. In Proceedings of the 20th ACM international conference on multimodal interaction. 2018: 400-408.
>
> [4] Xu Y, Liu GY. A method of emotion recognition based on ECG signal. In 2009 international conference on computational intelligence and natural computing. 2009, 1: 202-205.
>
> [5] Sarkar P, Etemad A. Self-supervised ECG representation learning for emotion recognition. IEEE Transactions on Affective Computing. 2020, 13(3): 1541-1554.
>
> [6] Zhu L, Spachos P, Ng PC, et. al. Stress detection through wrist-based electrodermal activity monitoring and machine learning. IEEE Journal of Biomedical and Health Informatics. 2023, 27(5): 2155-2165.

---

> ### Author Response · Authors · 2024-08-14
>
> Dear reviewer QG76,
>
> We appreciate your careful review and valuable comments on the manuscript. We attach great importance to your feedback and have started to revise and improve it. In addition, we have clarified some of your concerns.
>
> We understand that the discussion period is about to end, so we will remain online in the next period of time. If you have any questions or suggestions, we are very willing to communicate further and try our best to answer any concerns you have about the manuscript.
>
> Thank you again for your support and help in our work.
>
> Sincerely,
>
> Authors

---

> ### Comment · Area_Chair_AUSp · 2024-08-14
>
> Dear Reviewer QG76,
> Don't forget to engage in the conversation and letting the authors know about your take on their rebuttal before the deadline. Thanks for supporting NeurIPS.
> Best,

---

### Official Review · Reviewer_RxY3 · 2024-07-14

**Soundness:** 3
**Presentation:** 3
**Contribution:** 3
**Rating:** 6
**Confidence:** 3

**Summary:**

This paper proposes a robust multimodal sleep staging framework named Contrastive Imagination Modality Sleep Network (CIMSleepNet) to address the issues of missing modalities and temporal context modeling in automated sleep staging (ASS). CIMSleepNet combines a Modal Awareness Imagination Module (MAIM) for imputing missing data and a Semantic & Modal Calibration Contrastive Learning (SMCCL) approach to align the recovered data with real data distributions. Additionally, a multi-level cross-branch temporal attention mechanism is embedded to enhance the extraction of cross-scale temporal context representations. Extensive experiments on five multimodal sleep datasets demonstrate that CIMSleepNet significantly outperforms existing methods under various missing modality scenarios.

**Strengths:**

1. CIMSleepNet’s implementation of the Modal Awareness Imagination Module (MAIM) and Semantic & Modal Calibration Contrastive Learning (SMCCL) effectively addresses the issue of missing modalities. This robust recovery process aligns the recovered data distribution closely with the real data, significantly enhancing the performance and reliability of multimodal sleep staging under various incomplete modality conditions.
2. The framework’s multi-level cross-branch temporal attention mechanism (MCTA) excels in mining cross-scale temporal context representations at both intra-epoch and inter-epoch levels. This feature ensures a thorough and precise extraction of temporal features, which is crucial for accurate sleep staging.

**Weaknesses:**

Despite its robustness to missing modalities during inference, CIMSleepNet still requires a substantial amount of complete multimodal data for effective training. This dependence on complete data might pose challenges in real-world scenarios where obtaining comprehensive multimodal datasets is often difficult.

**Questions:**

The introduction of the Modal Awareness Imagination Module (MAIM) and the Semantic & Modal Calibration Contrastive Learning (SMCCL) adds considerable complexity to the CIMSleepNet framework. Could the authors provide a detailed analysis of the computational efficiency and resource requirements of CIMSleepNet compared to other state-of-the-art methods? Specifically, how does the increased computational overhead impact the scalability and real-time applicability of the framework in practical sleep monitoring systems?

**Limitations:**

see weakness

---

> ### Author Rebuttal · Authors · 2024-08-06
>
> **Weaknesses**:
>
> R1: Thank you for your valuable comment. In fact, our CIMSleepNet supports incomplete multimodal data in both the training and testing phases. Our proposed MAIM can recover the missing modality data from other available modality data of the same sample in incomplete multimodality data. Then, the semantic and modality information of the recovered data are corrected by the proposed SMCCL to generate data that matches the true distribution. As reported in Table 1 from the original manuscript, we take the Sleep-EDF20 dataset (containing two modalities) as an example. When the missing rate reaches 0.5, all multimodal samples in the training set are incomplete, but CIMSleepNet still achieves a high degree of robustness.
>
> **Questions**:
>
> R1: Thank you for your suggestion! We further analyze the computational efficiency and resource requirements of CIMSleepNet. Please refer to the Author Rebuttal for details.

---

> ### Author Response · Authors · 2024-08-14
>
> Dear reviewer RxY3,
>
> We appreciate your careful review and valuable comments on the manuscript. We attach great importance to your feedback and have started to revise and improve it. In addition, we have clarified some of your concerns.
>
> We understand that the discussion period is about to end, so we will remain online in the next period of time. If you have any questions or suggestions, we are very willing to communicate further and try our best to answer any concerns you have about the manuscript.
>
> Thank you again for your support and help in our work.
>
> Sincerely,
>
> Authors

---

> ### Comment · Area_Chair_AUSp · 2024-08-14
>
> Dear Reviewer RxY3,
> Don't forget to engage in the conversation and letting the authors know about your take on their rebuttal before the deadline. Thanks for supporting NeurIPS.
> Best,

---

### Official Review · Reviewer_PgT4 · 2024-07-30

**Soundness:** 2
**Presentation:** 3
**Contribution:** 2
**Rating:** 4
**Confidence:** 4

**Summary:**

The paper introduces a framework designed for sleep staging that addresses challenges associated with missing modalities in multimodal physiological signal datasets. The proposed model incorporates a modal awareness module and a semantic & modal calibration contrastive learning mechanism to handle missing data and ensure semantic consistency across modalities.

**Strengths:**

- the integration of MAIM and SMCCL to address the missing modality challenge could be useful for real-world scenarios where data incompleteness is common.
- the utilization of a multi-level cross-branch temporal attention mechanism enables the model to capture both intra-epoch and inter-epoch temporal dependencies effectively.
- the model is thoroughly evaluated on five sleep datasets.
- providing the code in supplementary materials fosters transparency and allows for community-driven improvements and testing in diverse scenarios.

**Weaknesses:**

- the paper introduces a potentially high computational overhead due to the complexity of the proposed model, especially with the integration of components like MAIM, SMCCL, and the multi-level attention mechanism.
- while the model performs well on controlled datasets, the real-world effectiveness and adaptability of the model in handling various types and degrees of missing data in uncontrolled environments are not extensively discussed.
- the paper does not extensively compare the proposed model with other state-of-the-art approaches that use less computationally intensive methods to handle missing modalities, which could provide a better balance between performance and efficiency.
- the method's reliance on existing multimodal data and potentially biased toward the modalities and specific characteristics of the datasets used for training could limit its effectiveness across broader populations or different types of sleep-related conditions.

**Questions:**

The main issue with the paper is the idea of trying to prove the concept of missing modality is very important. The idea could simply be circumvented by training a model for the modalities available and when all modalities are available, this approach is too complex compared to other approaches that are simpler. There are better sleep staging approaches which have not been compared against.

---

> ### Author Rebuttal · Authors · 2024-08-06
>
> **Weaknesses**:
>
> R1: Please see **Author Rebuttal** for details.
>
> R2: In response to this issue, we would like to clarify and add the following: Firstly, we conduct a comprehensive validation of the CIMSleepNet's effectiveness under random partial missing and complete missing conditions across five different multimodal sleep datasets. Meanwhile, we also explore the impact of different missing rates on the CIMSleepNet and set five different missing situations at each missing rate. These preset missing conditions cover a variety of possible practical application scenarios, thereby effectively simulating data missing conditions in natural environments. Furthermore, these experimental strategies are consistent with previous work on incomplete multimodal learning [1][2][3][4], which is the only way to evaluate the performance difference of the model under the same dataset with complete and incomplete modalities.
>
> R3: Thank you for your valuable comment. In the field of sleep staging, there is currently only one work on incomplete multimodal learning [1]. We have compared with this state-of-the-art method and indicated that our CIMSleepNet performs better in handling missing modalities. Although there are works [3][4] in other fields to deal with missing modalities, most of them require the modality to be complete in the training set and are not applicable to the field of sleep staging. In our study, the multimodal data in the training set is also incomplete, which makes our study closer to practical applications. Moreover, we also compare two non-sleep staging methods related to our SMCCL to deal with missing modalities, i.e., invariant contrastive learning [5] and semantic contrastive learning [2]. The results further exhibit the effectiveness of our core components. These comparative experiments demonstrate the advantages of our method in dealing with missing modal data. As for the issue of computational efficiency, as described in the Author Rebuttal, the model size and computational time of our method are within an acceptable range.
>
> R4: We understand your concerns regarding the model's generalization ability for sleep staging across different modalities. However, current multimodal sleep staging methods mainly rely on EEG, EOG, and EMG signals, which are also the data used by most previous sleep staging studies and our study. Moreover, we apply our method to heart rate and motion signals collected from real wearable devices and verify the generalization ability of the model across different modalities. In fact, the multimodal dataset used in this study already covers the main physiological signals commonly used in current sleep staging study [6]. These signals have been widely verified and recognized, which can be effectively used for sleep staging.
>
> **Questions**:
>
> R1: Thank you for your valuable comment. We actually deal with the issue of incomplete modalities in both the training and testing sets, which cannot be solved simply by training a model on the available modalities. On the contrary, our method is more suitable for real application scenarios. Because in practical applications, the signals collected by devices, especially portable wearable devices, often encounter incomplete modal data. Hence, each component we designed is necessary in this context. Furthermore, we demonstrate in Author Rebuttal that the complexity of our model is within an acceptable range. Currently, in the field of sleep staging, there is only one work that involves the problem of multimodal missingness, which we have compared in our paper. In addition, we also compare some advanced multimodal sleep staging methods with open source code and test the performance of these methods in the case of incomplete multimodality. Some advanced sleep staging methods are not compared because some of them only support single-modality sleep staging. There are several new multimodal sleep staging methods [7][8], but since their codes are not open source, we cannot reproduce these methods to test their performance in the absence of modality. Subsequently, we will present the results of these methods as reported in their original papers. Their performance is similar to that of our method on the complete multimodaliies.
>
>
>
> [1] Kontras K, Chatzichristos C, Phan H, et. al. CoRe-Sleep: A Multimodal Fusion Framework for Time Series Robust to Imperfect Modalities. IEEE Transactions on Neural Systems and Rehabilitation Engineering. 2024, 32: 840-849.
>
> [2] Qian S, Wang C. COM: Contrastive Masked-attention model for incomplete multimodal learning. Neural Networks. 2023, 162: 443-455.
>
> [3] Wang Y, Li Y, Cui Z. Incomplete multimodality-diffused emotion recognition. In Proceedings of the 37th International Conference on Neural Information Processing Systems. 2023: 17117-17128.
>
> [4] Lian Z, Chen L, Sun L, Liu B, Tao J. GCNet: Graph completion network for incomplete multimodal learning in conversation. IEEE Transactions on pattern analysis and machine intelligence. 2023, 45(7): 8419-8432.
>
> [5] Liu R, Zuo H, Lian Z, et. al. Contrastive Learning based Modality-Invariant Feature Acquisition for Robust Multimodal Emotion Recognition with Missing Modalities. IEEE Transactions on Affective Computing. 2024: 1-18.
>
> [6] Faust O, Razaghi H, Barika R, et. al. A review of automated sleep stage scoring based on physiological signals for the new millennia. Computer methods and programs in biomedicine. 2019, 176: 81-91.
>
> [7] Zhu H, Zhou W, Fu C, et. al. Masksleepnet: A cross-modality adaptation neural network for heterogeneous signals processing in sleep staging. IEEE Journal of Biomedical and Health Informatics. 2023, 27(5): 2353-2364.
>
> [8] Li T, Gong Y, Lv Y, et. al. GAC-SleepNet: A dual-structured sleep staging method based on graph structure and Euclidean structure. Computers in Biology and Medicine. 2023, 165: 107477.

---

> ### Author Response · Authors · 2024-08-12
>
> Dear Reviewer PgT4,
>
> I would like to thank you again for your time and effort in reviewing this work and for your insightful comments. We have responded to your comments and questions in detail in our rebuttal, especially your concerns about the complexity of the method for dealing with missing modalities. The experimental results show that our method is within an acceptable range in terms of complexity and has good real-time performance.
>
> Furthermore, regarding your question "The idea could simply be circumvented by training a model for the modalities available.", we would like to further clarify from two aspects. The details are as follows:
>
> 1) Due to the limitations of the acquisition equipment and the uncontrollable factors of the subjects during sleep, it is difficult to ensure the completeness of all modal data. Hence, it is unrealistic to solve the problem of modal missing in the test set by training with complete modal data and simulating modal missing conditions, such as the studies [1][2][3]. On the contrary, our method supports incomplete modal data in both the training set and the test set, which is more in line with practical applications.
>
> 2) When data is missing, it is not feasible to simply use other available data from the same modality for model training. This is because sleep data consists of continuous time signals throughout the night, with a high degree of temporal information embedded among samples [4][5]. When there are missing modalities, simply using available modality data would disrupt the temporal dependencies across samples.
>
> If you have any further questions or need additional clarifications, please do not hesitate to reach out. I am eager to ensure that the paper meets the highest possible standards and would value any additional feedback you might have.
>
> Thank you again for your time and consideration.
>
> Best regards,
>
> Authors
>
> [1] Lin Y, Gou Y, Liu X, et. al. Dual contrastive prediction for incomplete multi-view representation learning. IEEE Transactions on Pattern Analysis and Machine Intelligence. 2022, 45(4): 4447-4461.
>
> [2] Wang Y, Li Y, Cui Z. Incomplete multimodality-diffused emotion recognition. In Proceedings of the 37th International Conference on Neural Information Processing Systems. 2023: 17117-17128.
>
> [3] Sun L, Lian Z, Liu B, et. al. Efficient multimodal transformer with dual-level feature restoration for robust multimodal sentiment analysis. IEEE Transactions on Affective Computing. 2023, 15(1): 309-325.
>
> [4] Supratak A, Dong H, Wu C, et. al. DeepSleepNet: A model for automatic sleep stage scoring based on raw single-channel EEG. IEEE transactions on neural systems and rehabilitation engineering. 2017, 25(11): 1998-2008.
>
> [5] Phyo J, Ko W, Jeon E, et. al. TransSleep: Transitioning-aware attention-based deep neural network for sleep staging. IEEE Transactions on Cybernetics. 2022, 53(7): 4500-4510.

---

> > ### Author Response · Authors · 2024-08-14
> >
> > Dear reviewer PgT4,
> >
> > We appreciate your careful review and valuable comments on the manuscript. We attach great importance to your feedback and have started to revise and improve it. In addition, we have clarified some of your concerns.
> >
> > We understand that the discussion period is about to end, so we will remain online in the next period of time. If you have any questions or suggestions, we are very willing to communicate further and try our best to answer any concerns you have about the manuscript.
> >
> > Thank you again for your support and help in our work.
> >
> > Sincerely,
> >
> > Authors

---

> ### Comment · Area_Chair_AUSp · 2024-08-14
>
> Dear Reviewer PgT4,
> Don't forget to engage in the conversation and letting the authors know about your take on their rebuttal before the deadline. Thanks for supporting NeurIPS.
> Best,

---

### Author Rebuttal · Authors · 2024-08-06

Thank you for your decision and constructive comments on our manuscript. We have tried our best to modify some of the content and uploaded a PDF consisting of figures and tables. In the subsequent rebuttal per review, we will refer to it as "PDF". The blue font part in the PDF is the newly added content. Furthermore, we have also made detailed clarifications for some of the questions and misunderstandings that may exist in our manuscript in the comments. We have analyzed the computational efficiency issues that most reviewers are concerned about in this section. For other responses, please see the rebuttal per review for detailed responses.

**Analysis of the Computational Efficiency and Resource Requirements**

As indicated in Table 1 from the PDF, we calculate the parameters, FLOPs (single inference), and average training time per iteration of CIMSleepNet and other state-of-the-art methods on the Sleep-EDF-20 dataset. Moreover, we also deploy the weights of each model on the Raspberry Pi 4B platform to test its inference time, thereby simulating its running efficiency on the mobile device. From Table 1, we observe that the sequence modeling models (i.e., TransSleep, XSleepNet, and our CIMSleepNet) are still within acceptable ranges despite having higher model size and computation time than non-sequence modeling models. Note that, sequence modeling refers to learning temporal context information using recurrent neural networks and Transformers, etc. Because all models can be successfully deployed on the Raspberry Pi 4B platform, and the average inference time of these methods for a single data epoch (duration of 30 s) is between 0.3 s and 1.1 s, which is much lower than 30 s. This means that all models exhibit good real-time performance, and after processing the current data epoch, there is enough time to process the data of the next epoch.

As shown in Table 2 from the PDF, we further analyze the memory consumption and computational complexity of each component in CIMSleepNet. We find that although the two components designed to mitigate modality missing issue (MAIM and SMCCL) introduce additional parameters and FLOPs, the increase is much less than that introduced by the sequence modeling component (MCTA). This phenomenon is similar to what is observed in TransSleep and XSleepNet, where the use of sequence modeling methods also leads to a significant increase in the parameters. However, compared to non-sequence modeling methods, sequence modeling methods can demonstrate more powerful potential on large-scale sleep datasets [1][2]. As reported in Table 3 from the PDF, we train and test all the methods in Table 1 on a large-scale dataset, i.e., the SHHS dataset. The results show that the performance of non-sequence modeling methods on large-scale datasets is far inferior to that of sequence modeling methods. Although SalientSleepNet and MM-Net improve performance on small-scale datasets by enhancing the structure of convolutional neural networks (CNNs), these CNN improvements are far less effective than sequence modeling methods on large-scale datasets. Hence, we abandon the improvement of CNN structure and focus on optimizing sequence modeling methods to achieve excellent performance on both small-scale and large-scale datasets.

[1] Phan H, Lorenzen KP, Heremans E, et. al. L-SeqSleepNet: Whole-cycle long sequence modelling for automatic sleep staging. IEEE Journal of Biomedical and Health Informatics. 2023, 27(10): 4748-4757.

[2] Phan H, Mikkelsen K, Chén OY, et. al. Sleeptransformer: Automatic sleep staging with interpretability and uncertainty quantification. IEEE Transactions on Biomedical Engineering. 2022, 69(8): 2456-2467.

---

### Decision · Program_Chairs · 2024-09-25

**Decision:**

Accept (poster)

**Comment:**

This decision was perhaps the most difficult one, not only because of the larger spread of scores but also because the overall recommendation would be approval but I can see myself more inclined with the lower score. Moreover, both the reviews and the discussion were appalling despite my encouragement to the reviewers to engage. Hence, applying the in dubio pro reo principle and respecting the majority I opted for recommending acceptance as poster.
Regarding the paper itself, the reviews (weak as they look to me) showed praises on the preliminaries i.e. problem statement and justification (R-HATA), and the extensive experimentation and thorough evaluation (R-RxY3 and R-PgT4) across many datasets (R-QG76). Counterbalancing this strengths, reviewers mainly showed consensus on two things; the computational complexity and cost (R-QG76 and R-RxY3, as well as R-HATA which uses the related term “mathematically nuanced”) and its difficult in translating to the real world (R-RxY3). During the rebuttal, the authors made an attempt at addressing the former i.e. “we further analyze the memory consumption and computational complexity of each component”, but did not appear to address the later.
Anyhow, in its current status, I can see the potential of the work but I can clearly see why the scores were not unanimously enthusiastic.
No issues with ethics were highlighted.